# SAMPLE LOTTERY: UNSUPERVISED DISCOVERY OF CRITICAL INSTANCES IN RLVR OF LLMS

**Zhiping Xiao**[1][*], **Yusheng Zhao**[2][*], **Qixin Zhang**[3][*], **Jiaye Xie**[2][†], **Wanjia Zhao**[4],
**Weizhi Zhang**[5], **Xiao Luo**[6][†], **Philip S. Yu**[5], **Ming Zhang**[2][†]

[1] Paul G. Allen School of Computer Science and Engineering, University of Washington
[2] State Key Laboratory for Multimedia Information Processing,
School of Computer Science, PKU-Anker LLM Lab, Peking University
[3] College of Computing and Data Science, Nanyang Technological University
[4] Department of Computer Science, Stanford University,
[5] Department of Computer Science, University of Illinois Chicago,
[6] Department of Statistics, University of Wisconsin–Madison

`patxiao@uw.edu,yushengzhao99@gmail.com,qixinzhang1106@gmail.com`
`jyyeahofficial@gmail.com,wanjiazh@stanford.edu,`
`{wzhan42,psyu}@uic.edu,xiao.luo@wisc.edu,mzhang_cs@pku.edu.cn`

## ABSTRACT

Reinforcement Learning with Verifiable Reward (RLVR) has equipped large language models (LLMs) with the capability of reasoning over complicated logical problems through policy optimization. However, conventional methods require complete annotation of the entire dataset and allocate computation resources uniformly over all samples. We articulate the *lottery sample hypothesis* in policy optimization of LLMs: a large training set contains a small subset that, when trained alone, yields performance comparable to that of the full dataset. This paper therefore explores the following question: ***How can we identify these lottery-winning samples from the original dataset without access to answers?*** Unlike those prior efforts that analyze the effect of different samples in the training set with complete annotation, this paper focuses on the unsupervised discovery of critical instances for LLM reasoning and proposes a novel framework termed Complementary Conformal Selection (`CONST`). Specifically, `CONST` evaluates the importance of samples by considering two complementary components: *procedural volatility* and *outcome volatility*. Procedural volatility measures the potential variations during the LLM's reasoning process, while outcome volatility captures inconsistencies in the final answer. Subsequently, conformal prediction is used to obtain a prediction set whose cardinality serves as the criterion for selecting the lottery-winning samples for annotation. We also provide a theoretical analysis, showing that `CONST` can effectively approximate the optimal policy. Extensive experiments on various LLMs across different datasets demonstrate that `CONST` is annotation-efficient, high-performing and model-agnostic. The code is available at https://github.com/YushengZhao/SampleLottery.

## 1 INTRODUCTION

Reinforcement learning (RL) has recently become an essential tool for post-training of large language models (LLMs) (Anil et al., 2023; OpenAI, 2024; 2025; Guo et al., 2025; Du et al., 2025). Policy optimization algorithms (Schulman et al., 2017; Shao et al., 2024) have significantly enhanced the logical reasoning capabilities of LLMs (DeepMind, 2024; Wang et al., 2025a; Ren et al., 2025). For logical problems, directly verifiable answers provide straightforward rewards for reinforcement learning, enabling effective outcome supervision of LLMs. This approach, known as reinforcement learning with verifiable reward (RLVR) (Gao et al., 2024; Lambert et al., 2024), is commonly im-

---

[*]Equal contribution.
[†]Corresponding authors.

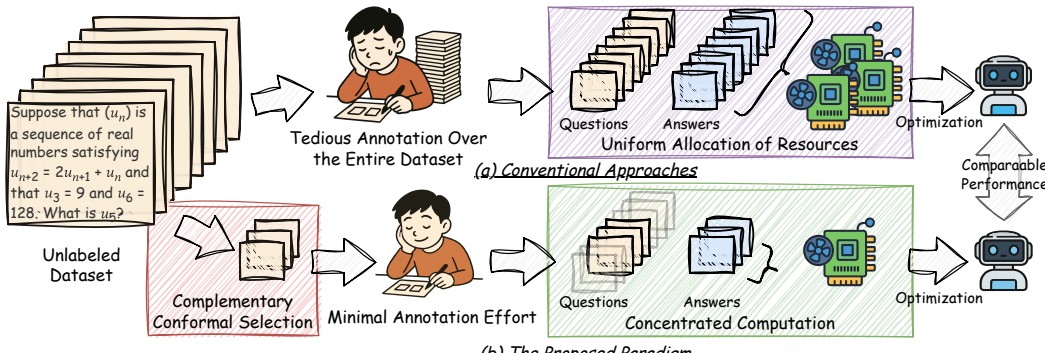

Figure 1: Conventional approaches require tedious full annotation over the entire dataset and allocate computation resources uniformly across the training set. By comparison, this work selects lottery-winning samples from the training set in an unsupervised manner, and then optimizes the model using these critical instances only, achieving comparable performance.

plemented using algorithms such as Group Relative Policy Optimization (GRPO) (Shao et al., 2024) and its variants (Liu et al., 2025d; Chen et al., 2025; Pang & Jin, 2025; Zhang et al., 2025b).

Despite the significant improvement, conventional approaches (Hao et al., 2025; Wu et al., 2025; Di et al., 2025) demand full annotation over the entire dataset for verification, and often allocate computation resources uniformly across the full dataset. Nevertheless, some recent studies in RLVR of LLMs (Chen et al., 2025; Wang et al., 2025c; Vanlioglu, 2025) and many prior works on the more general field of data subset selection and valuation (Paul et al., 2021; Das et al., 2021b; Guo et al., 2022; Das et al., 2024) suggest that the instances in the training set are not equally important, and that training on a small subset may also lead to satisfactory results (Wang et al., 2025c). Based on these findings, we articulate the *lottery sample hypothesis* in RLVR of LLMs:

> *A large training set for RLVR on LLMs contains a small subset that, when trained alone, can achieve performance comparable to that of the full dataset.*

With this hypothesis, it is possible to break conventional approaches from two aspects (as illustrated in Figure 1): *(i)* full annotation of the dataset is no longer required, and ground truth answers of several lottery-winning samples are sufficient; *(ii)* computation can be concentrated on several critical instances. Therefore, this paper explores a central question of this new paradigm:

> *How can we find the critical instances (the lottery-winning samples) for RLVR on LLMs from the original training set without annotation?*

To answer this question, this paper proposes a novel framework named Complementary Conformal Selection (CONST) for the unsupervised discovery of critical instances in the training set. CONST evaluates the value of each instance from two complementary perspectives: *procedural volatility* and *outcome volatility*. Procedural volatility assesses potential variations in reasoning chains by examining how different segments of reasoning affect the final answer. Outcome volatility measures inconsistencies in the final answers produced by different reasoning paths. Both yield multisets (*i.e.*, sets that allow duplicating elements) of results, which are then merged and fed into a conformal prediction module. The conformal prediction produces a prediction set, whose cardinality is used as the criterion for selecting lottery-winning samples. A theoretical analysis is also provided demonstrating that CONST can effectively approximate the optimal policy. We conduct extensive experiments across datasets with various LLMs, showing that CONST outperforms various baselines and enables comparable performance with $< 0.5\%$ of the samples. Our contribution is summarized as follows:

❶ *New Perspective*: We present a probabilistic perspective for the unsupervised identification of critical instances in the full dataset for further annotation and RLVR optimization on LLMs, enabling an annotation-minimal, data-efficient and performance-competitive optimization procedure compared with training on the entire fully annotated dataset.

❷ *Novel Methodology with Theoretical Analysis*: We propose CONST, a probabilistic approach based on conformal prediction that considers both procedural volatility and outcome volatility in

LLM reasoning, to select lottery-winning samples for annotation and optimization. Notably, we provide a rigorous theoretical analysis of `CONST` demonstrating that it can effectively approximate the optimal policy parameter setup under the lottery sample hypothesis.

❸ *Empirical Validation*: We conduct extensive experiments across four mathematical datasets on various LLMs against competing baselines, showing that `CONST` is **(1) annotation-efficient**, achieving similar performance to the full dataset with $< 0.5\%$ of the annotation, **(2) high-performing**, outperforming competitive baselines by $10.97\%$ on average, and **(3) model-agnostic**, showing consistent improvement across three different architectures.

## 2 PRELIMINARIES

**Problem Definition.** Given a training set of questions $\mathcal{Q} = \{X_1, X_2, \ldots, X_N\}$ from the input space $\mathcal{X}$, conventional RLVR approaches first annotate the dataset with ground truth answers $\mathcal{A} = \{Y_1, Y_2, \ldots, Y_N\}$ in the output space $\mathcal{Y}$, and then optimize the original LLM (*i.e.*, the policy) $\pi_0$ with RL algorithms (*e.g.*, GRPO), *i.e.*, $\pi^F = \Phi(\pi_0, \mathcal{Q}, \mathcal{A})$, where $\pi^F$ is the policy optimized with full data annotation and $\Phi(\cdot, \cdot, \cdot)$ is the optimization process of RLVR. Our goal is to find a subset of $\mathcal{Q}$ with budget $b$, *i.e.*, $\mathcal{Q}' \subset \mathcal{Q}$ and $|\mathcal{Q}'| = b$, and then annotate the selected data with answers $\mathcal{A}'$ so that the optimized policy $\pi^P = \Phi(\pi_0, \mathcal{Q}', \mathcal{A}')$ achieves comparable performance to $\pi^F$.

**Reinforcement Learning with Verifiable Reward.** During the training process of reinforcement learning with verifiable reward, the LLM generates a list of $n$ outputs $\{O_1, O_2, \ldots, O_n\}$ for a question $X$, and the outputs are verified against the ground truth answer $Y$ to obtain the rewards $r_1, r_2, \ldots, r_n$, where correct answers receive 1 and incorrect ones 0. In the widely-adopted group relative policy optimization (Shao et al., 2024; Liu et al., 2025c; Guo et al., 2025), the advantage function of each output $O_i$ is computed as:

$$a_i = \frac{r_i - \text{mean}(\{r_j\}_{j=1}^n)}{\text{std}(\{r_j\}_{j=1}^n)}. \tag{1}$$

With this, the GRPO optimization objective can be formulated as follows:

$$\mathcal{L}_{\text{GRPO}} = \mathbb{E}_{O_i \sim \pi_{\theta'}} \left[ -\frac{1}{n} \sum_{i=1}^n \left( \min \left( \frac{\pi_\theta(O_i|X)}{\pi_{\theta'}(O_i|X)} a_i, \text{clip}\left( \frac{\pi_\theta(O_i|X)}{\pi_{\theta'}(O_i|X)}, 1 - \varepsilon, 1 + \varepsilon \right) a_i \right) \right) \right] \tag{2}$$
$$+ \beta \mathcal{D}_{\text{KL}}(\pi_\theta || \pi_0),$$

where $\pi_\theta$ is the current policy, $\pi_{\theta'}$ is the old policy, and $\pi_0$ is the reference policy.

**Conformal Prediction.** Conformal prediction (CP) (Vovk et al., 2005; Papadopoulos et al., 2007; Angelopoulos et al., 2023; Barber et al., 2023; Su et al., 2024) is a model-agnostic solution to obtain prediction sets (or intervals in regression tasks) that are mathematically guaranteed to cover the ground truth answer with high probabilities. Concretely, given an input from $\mathcal{X}$, a prediction from $\mathcal{Y}$ and the prediction model $\pi(Y|X)$, a scoring function $f^\pi : \mathcal{X} \times \mathcal{Y} \to \mathbb{R}$ is defined to quantify the disagreement between the input and the predicted answer. The scoring function is applied to a calibration set $\mathcal{D}^{\text{cal}} = \{(X_i^{\text{cal}}, Y_i^{\text{cal}})\}_{i=1}^m$, which is independently and identically distributed as the dataset under consideration, to obtain the calibration scores $f^\pi(X_i^{\text{cal}}, Y_i^{\text{cal}}), i = 1, 2, \ldots, m$. For a given confidence level $1 - \alpha$, a threshold $\rho$ can be determined by taking the $\frac{\lceil (m+1)(1-\alpha) \rceil}{m}$ quantile of these scores. For a given input $X \in \mathcal{X}$, and the predefined error rate $\alpha$, the conformal prediction set is guaranteed to cover the correct answer with a probability of $1 - \alpha$, defined as follows:

$$\mathcal{C}_{1-\alpha}(X) = \{Y \mid f^\pi(X, Y) \leq \rho\}. \tag{3}$$

## 3 THE PROPOSED `CONST`

### 3.1 FRAMEWORK OVERVIEW

The overall idea of this work is to first select the critical instances $\mathcal{Q}'$ from the training set of questions $\mathcal{Q}$ using the proposed Complementary Conformal Prediction (`CONST`), then annotate these critical instances with answers $\mathcal{A}'$, and finally optimize the LLM using off-the-shelf RLVR algorithms

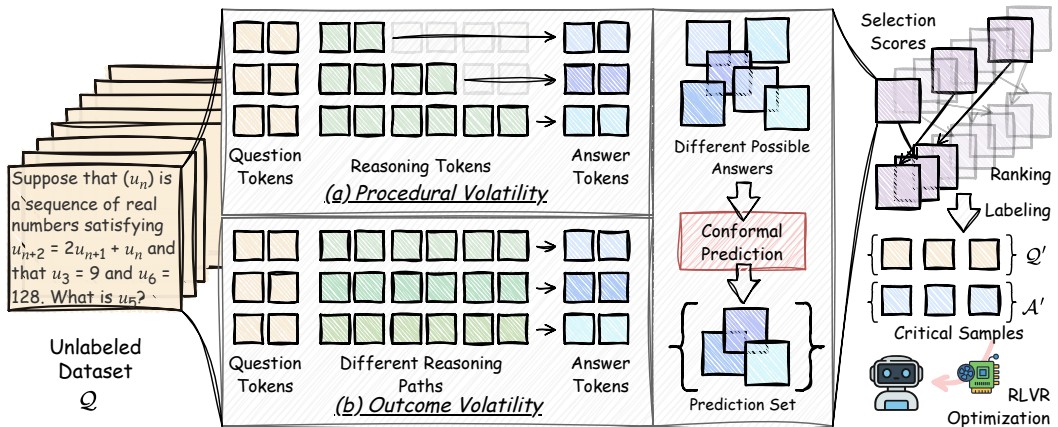

Figure 2: The overall framework of the proposed CONST. CONST selects critical samples via conformal prediction based on both procedural volatility and outcome volatility. The selected samples are then annotated with ground truth answers for standard RLVR optimization on LLMs.

such as GRPO. The proposed CONST evaluates the importance of each instance $X \in \mathcal{Q}$ from two perspectives: procedural volatility and outcome volatility. Procedural volatility measures how the final answer to the input question changes when the reasoning paths are truncated at different stages, while outcome volatility captures the inconsistencies in the final answers produced by different reasoning paths. Conformal prediction is then used to obtain a prediction set for each instance, and the cardinality of these sets serves as the criterion for selecting the critical instances. We illustrate the overall framework in Figure 2.

## 3.2 PROCEDURAL VOLATILITY

Questions important for policy optimization are expected to induce reasoning trajectories of significant complexity, as straightforward thinking chains are less likely to substantially enhance the model's logical reasoning abilities. Therefore, we propose measuring the volatility in the reasoning process by truncating the thinking chains at different stages and prompting the LLM to provide the final answers directly. Simple and straightforward reasoning paths are more likely to yield consistent answers, whereas complex, intricate ones are more likely to exhibit volatility.

Specifically, given a question $X \in \mathcal{Q}$, we first deterministically sample an output, denoted as $O = [t_1, t_2, ..., t_L; \widehat{Y}] = \pi_0(O|X)$, where $t_1, t_2, \ldots, t_L$ are tokens of the reasoning path and $\widehat{Y}$ is the predicted answer typically enclosed in special formats like \box{}. We truncate the reasoning path at different stages to obtain a set of truncated reasoning trajectories, defined as follows:

$$\mathcal{T}(X) = \{[t_1, t_2, \ldots, t_{\lceil \frac{iL}{n_P} \rceil}] \mid i = 1, 2, \ldots, n_P\}, \tag{4}$$

where $n_P$ is the number of stages. Subsequently, we query the LLM with these truncated trajectories and ask the model to directly output a single final answer for each truncated trajectory without reasoning. Formally, we obtain a multiset (bag) [1] for each sample $X \in \mathcal{Q}$ through response truncation and LLM re-querying:

$$\mathcal{B}^P(X) = \{\{\widehat{Y} = \pi_0(\widehat{Y}|X, \tau) \mid \tau \in \mathcal{T}(X)\}\}. \tag{5}$$

## 3.3 OUTCOME VOLATILITY

In addition to procedural volatility, we also consider variations in the final answers. During policy optimization, diverse answers to a question induce gradients from multiple directions, helping the model avoid pitfalls from different sources. To quantify this diversity, we introduce outcome

---

[1] Different from sets, multisets (bags) allow repetition of elements, with each element associated with a count of its appearance. For convenience, we use $|\cdot|$ to denote the size of the multiset, defined as the sum of the counts of all elements, and $\uplus$ to denote the union of multisets, where the counts of each element are added.

volatility, which evaluates the inconsistencies in the final answers (outcomes) produced by different possible reasoning trajectories sampled from the policy.

In particular, given the original policy (the LLM before RLVR training) $\pi_0$ and a question $X \in \mathcal{Q}$ under consideration, we directly sample $n_O$ outputs from the policy to obtain a multiset, *i.e.*,

$$\mathcal{B}^O(X) = \{\{\widehat{Y}_i \mid i = 1, 2, \ldots, n_O, \widehat{Y}_i \sim \pi_0(\widehat{Y}|X)\}\}. \tag{6}$$

### 3.4 Conformal Prediction

Procedural volatility and outcome volatility generate multisets for each instance, encompassing the possible answers of LLMs. We are now concerned with how many of these answers are *likely* to be the correct answer. To address this, we employ conformal prediction (Vovk et al., 2005; Su et al., 2024) as a theoretically grounded solution. The overall procedures of computing conformal prediction sets have been discussed in Section 2, and in the following, we will present the design of the scoring function $f$ and how the prediction sets $\mathcal{C}_{1-\alpha}(X), X \in \mathcal{Q}$ are obtained.

As discussed in Section 2, the scoring function $f^{\pi_0}(X, Y) \in \mathbb{R}$ is designed to quantify the disagreement between the input question $X$ and the final answer $Y$. In other words, when the model $\pi_0(Y|X)$ is certain that $Y$ is the correct answer given the input $X$, $f^{\pi_0}(X, Y)$ will be low, and vice versa. Conformal prediction does not require the scoring function to have theoretical guarantees of the measurement of certainty, although a good measurement is preferred. For each input question $X \in \mathcal{Q}$ and a predicted answer $\widehat{Y} \in \mathcal{B}(X) = \mathcal{B}^P(X) \uplus \mathcal{B}^O(X)$, we compute the scoring function by comparing $\widehat{Y}$ with other elements in $\mathcal{B}(X)$. Specifically, as consistent predictions are a natural sign of certainty (Wang et al., 2022), we use the negative frequency of $\widehat{Y}$ in $\mathcal{B}(X)$, *i.e.*,

$$f_{\text{NF}}(X, \widehat{Y}) = -\text{freq}(\widehat{Y}; \mathcal{B}(X)) = -\frac{\text{count}_{\mathcal{B}(X)}(\widehat{Y})}{|\mathcal{B}(X)|}, \tag{7}$$

where $\text{count}_{\mathcal{B}(X)}(\cdot)$ returns the number of elements in the multiset. Nevertheless, using negative frequency alone as the scoring function may cause the scores to be concentrated on certain values, and therefore, entropy is used for fine-grained measurement, defined as follows:

$$f_{\text{ent}}(X, \widehat{Y}) = \frac{H(\mathcal{B}(X))}{\log |\mathcal{B}(X)|} = \frac{-\sum_{Y' \in \text{set}(\mathcal{B}(X))} \text{freq}(Y'; \mathcal{B}(X)) \log \text{freq}(Y'; \mathcal{B}(X))}{\log |\mathcal{B}(X)|}, \tag{8}$$

where $\text{set}(\cdot)$ returns the non-repeating elements in the multiset. Finally, the negative frequency and entropic scores are combined to obtain the final scoring function:

$$f^{\pi_0}(X, \widehat{Y}) = f_{\text{NF}}(X, \widehat{Y}) + \lambda \cdot f_{\text{ent}}(X, \widehat{Y}), \tag{9}$$

where $\lambda$ is a hyperparameter balancing the two terms. With the scoring function, we then calibrate it with a calibration set as described in Section 2 to obtain the threshold $\widehat{\rho}$. Note that when computing the calibration scores, if the correct answer $Y_i^{\text{cal}}$ does not appear in the final multiset $\mathcal{B}(X_i^{\text{cal}})$, we set the score $f(X_i^{\text{cal}}, Y_i^{\text{cal}})$ to $+\infty$. Thus, we obtain the conformal prediction set for each $X \in \mathcal{Q}$ as:

$$\widehat{\mathcal{C}}_{1-\alpha}(X) = \{\widehat{Y} \in \text{set}(\mathcal{B}(X)) \mid f^{\pi_0}(X, \widehat{Y}) \leq \widehat{\rho}\}. \tag{10}$$

### 3.5 Model Optimization

The size of the conformal prediction set naturally measures how many of the answers that the model considers likely to be correct. A larger size indicates richer and more effective optimization signals associated with the correct answer during model training. Therefore, we use the size of the conformal prediction set as the criterion for selecting critical samples. Additionally, to encourage sample diversity, we cluster the set of questions $\mathcal{Q}$ into $b$ groups $\mathcal{Q}_1, \mathcal{Q}_2, \ldots, \mathcal{Q}_b$ before selecting the samples from each group. Formally, the selection process can be written as follows:

$$\mathcal{Q}' = \left\{ \arg\max_{X \in \mathcal{Q}_i} |\widehat{\mathcal{C}}_{1-\alpha}(X)| \mid i = 1, 2, \ldots, b \right\}. \tag{11}$$

With the selected set of questions $\mathcal{Q}' \subset \mathcal{Q}$, we annotate this small subset to obtain its ground truth answers $\mathcal{A}'$. Finally, we use the standard RLVR algorithm (*i.e.*, GRPO), as described in Section 2, to optimize the model $\pi_0$ with $\mathcal{Q}'$ and $\mathcal{A}'$ to obtain $\pi^P$ as our final model. A summary of the execution pipeline of the proposed **CONST** is presented in Algorithm 1.

---

**Algorithm 1:** The execution pipeline of `CONST`

---

**Input:** The set of questions $\mathcal{Q}$, the original policy $\pi_0$, the calibration set $\mathcal{D}^{\text{cal}} = \{(X_i^{\text{cal}}, Y_i^{\text{cal}})\}_{i=1}^m$

1  Initialize `CalScoreList` and `SizeList` as empty lists

2  **for** iteration $i \leftarrow 1$ **to** $m$ **do**                          `// Calibrate the scoring function`

3       Calculate the score $f^{\pi_0}(X_i^{\text{cal}}, Y_i^{\text{cal}})$ with Eq. 9 and append it to `CalScoreList`

4  **end**

5  Find the $\frac{\lceil (m+1)(1-\alpha) \rceil}{m}$ quantile of `CalScoreList` as $\widehat{\rho}$

6  **for** iteration $i \leftarrow 1$ **to** $N$ **do**                          `// Obtain the conformal prediction sets`

7       Calculate the multiset of possible answers $\mathcal{B}(X_i)$ according to Eq. 5 and Eq. 6

8       Calculate the score $f^{\pi_0}(X_i, \widehat{Y}_i)$ for each $\widehat{Y}_i \in \mathcal{B}(X_i)$ with Eq. 9

9       Obtain the prediction set $\widehat{\mathcal{C}}_{1-\alpha}(X)$ with $\widehat{\rho}$ using Eq. 10 and append the size of it to `SizeList`

10 **end**

11 Cluster $\mathcal{Q}$ into $b$ groups $\mathcal{Q}_1, \mathcal{Q}_2, \ldots, \mathcal{Q}_b$

12 Select the question with the largest size from each group to form $\mathcal{Q}'$    `// Select critical samples`

13 Annotate $\mathcal{Q}'$ with ground truth answers $\mathcal{A}'$                    `// Annotate several samples`

14 Optimize $\pi_0$ with $\mathcal{Q}'$ and $\mathcal{A}'$ to obtain the optimized model $\pi^P$  `// Optimize the policy using RL`

15 **return** $\pi^P$ as the optimized model

---

## 3.6 THEORETICAL ANALYSIS

Here, we aim to provide a theoretical understanding of our proposed `CONST` under the *lottery sample hypothesis*. Before going into the details, we first review the basic notions of ergodic Markov decision processes and mixing time (Puterman, 1990).

**Definition 3.1.** *A Markov decision process $\mathcal{M} \triangleq (\mathcal{S}, \mathcal{A}, P, r, \gamma)$ is ergodic if the induced Markov chain under **any** stationary policy admits a unique stationary distribution $\rho_\infty$. Moreover, the underlying Markov chain of an ergodic MDP is said to mix in time $t_{mix}(\epsilon)$ if*

$$t_{mix}(\epsilon) \triangleq \min\{t \geq 0 : \max_{s \in \mathcal{S}} \big( \max_{A \subseteq \mathcal{S}} (|\Pr(Y_t \in A | Y_0 = s) - \rho_\infty(A)|) \big) \leq \epsilon\},$$

*where $(Y_0, Y_1 \ldots, Y_t \ldots)$ denotes an induced Markov chain under any stationary policy and $\epsilon > 0$.*

**Remark 3.1.** *When $\epsilon < 1/2$, choosing a different $\epsilon$ only changes the mixing time up to a constant factor (Levin & Peres, 2017) and so one often fixes $\epsilon = 1/4$ and simply writes $t_{mix} \triangleq t_{mix}(1/4)$.*

With the concept of ergodic MDP, we next establish a generalization bound for our proposed `CONST` under the *lottery sample hypothesis*. Before doing so, we first introduce some frequently used notations. Specifically, since `CONST` only utilizes a subset of training instances $\mathcal{Q}' \subset \mathcal{Q}$ to optimize policy $\pi_\theta$ where $|\mathcal{Q}'| = b$. So, for any subset $S \subset \mathcal{Q}$, we define the empirical GRPO loss on $S$ as follows:

$$\hat{\mathcal{L}}_{\text{GRPO}}^S = -\frac{1}{n|S|} \sum_{X \in S} \sum_{i=1}^n \min \left( \frac{\pi_\theta(O_i|X)}{\pi_{\theta'}(O_i|X)} a_i, \text{clip}\big( \frac{\pi_\theta(O_i|X)}{\pi_{\theta'}(O_i|X)}, 1-\varepsilon, 1+\varepsilon \big) a_i \right) + \mathcal{D}_{\text{KL}}(\pi_\theta || \pi_0),$$

where $a_i \triangleq \frac{r_i - \text{mean}(\{r_j\}_{j \in S})}{\text{std}(\{r_j\}_{j \in S})}$ is the advantage calculated based on relative rewards. With this symbol, we then present the approximation assumption about the chosen subset $\mathcal{Q}'$, that is to say,

**Assumption 3.1** (Lottery Sample Hypothesis). *Subset $\mathcal{Q}'$ is said to be an $\epsilon$-approximation of the full training set $\mathcal{Q} \triangleq \{X_1, X_2, \ldots, X_N\}$ if $\|\nabla \hat{\mathcal{L}}_{GRPO}^{\mathcal{Q}'}(\theta) - \nabla \hat{\mathcal{L}}_{GRPO}^{\mathcal{Q}}(\theta)\|_2 \leq \epsilon$ holds for any parameter vector $\theta$ where the symbol $\| \cdot \|_2$ denotes $l_2$ norm and $\epsilon > 0$.*

Next, we make some standard assumptions in optimization theory (Li et al., 2018; Yue et al., 2023).

**Assumption 3.2** (Smoothness). *The objective $\hat{\mathcal{L}}_{GRPO}^Q(\theta)$ is L-smooth, that is, $\hat{\mathcal{L}}_{GRPO}^Q(\theta)$ is differentiable and there exists a constant $L > 0$ such that $\|\nabla \hat{\mathcal{L}}_{GRPO}^Q(x) - \nabla \hat{\mathcal{L}}_{GRPO}^Q(y)\|_2 \leq L\|x - y\|_2$.*

**Assumption 3.3** (Polyak-Łojasiewicz Condition). *There exists a constant $\mu > 0$ such that $2\mu\big(\hat{\mathcal{L}}_{GRPO}^Q(\theta) - \hat{\mathcal{L}}_{GRPO}^Q(\theta_{GRPO}^*)\big) \leq \|\nabla \hat{\mathcal{L}}_{GRPO}^Q(\theta)\|_2^2$ where $\theta_{GRPO}^* \triangleq \arg\min_\theta \hat{\mathcal{L}}_{GRPO}^Q(\theta)$.*

Furthermore, we suppose, at line 14 of Algorithm 1, the policy parameter vector $\theta$ is updated by standard gradient descent, i.e., $\theta_{k+1} \triangleq \theta_k - \frac{1}{L}\nabla\hat{\mathcal{L}}_{\text{GRPO}}^{\mathcal{Q}'}(\theta)$, where $L$ denotes the smoothness parameter. With all these preparations, we can have the following generalization theorem:

**Theorem 3.1** (Proof is deferred to Appendix A). *Under Assumption 3.1-3.3, if the underlying MDP $\mathcal{M} \triangleq (\mathcal{S}, \mathcal{A}, P, r, \gamma)$ is ergodic with mixed time $t_{mix}$ and the gradient $\nabla\hat{\mathcal{L}}_{GRPO}^{Q}(\theta)$ is bounded, i.e., $\|\nabla\hat{\mathcal{L}}_{GRPO}^{Q}(\theta)\|_2 \leq G$, then the following inequality holds with probability greater than $1 - \delta$, that is,*

$$\mathcal{L}_{GRPO}(\theta_k) - \mathcal{L}_{GRPO}(\theta^*) \leq 4\mathcal{R}(\mathcal{F}_{GR}) + \mathcal{O}\Big(\sqrt{\frac{t_{mix}\sigma_R^2(1-\frac{1}{n})\ln(\frac{1}{\delta})}{Nn}} + \frac{\ln(\frac{1}{\delta})}{Nn(1-\gamma)}\Big)$$

$$+ (1 - \frac{\mu}{L})^{k+1}\hat{\mathcal{L}}_{GRPO}^{Q}(\theta_0) + \frac{2G}{\mu}\epsilon + \frac{\epsilon^2}{2\mu},$$

*where $\theta^* \triangleq \arg\min_\theta \mathcal{L}_{GRPO}(\theta)$, $\mathcal{R}(\mathcal{F}_{GR})$ is the Rademacher complexity of the group-relative loss function space $\mathcal{F}_{GR}$, $N$ denotes the size of full training set $\mathcal{Q}$, $n$ is size of outputs for each question and $\sigma_R^2$ is an upper bound of variance of the return $\{r_i\}_{i=1}^n$, i.e., $Var_{\pi_\theta}(r_i) \leq \sigma_R^2, \forall\theta$.*

**Remark 3.2.** *Note that Rademacher complexity serves as a measure of the policy network's capacity to fit the training data, reflecting the richness of the function class it can represent. As a result, under the Lottery Sample Hypothesis, Theorem 3.1 implies that with sufficiently large question dataset and verified rewards, our proposed CONST can effectively approximate the optimal policy parameter $\theta^*$.*

## 4 EXPERIMENTS

### 4.1 EXPERIMENTAL SETUP

**Datasets and Evaluation.** During model training, we use the BigMath-sub dataset, a randomly selected subset containing 2048 instances of the BigMath dataset (Albalak et al., 2025). During the test phase, we use four mathematical reasoning datasets widely adopted in RLVR evaluation, i.e., AMC 23 (problems & solutions, 2023), MinervaMath (Lewkowycz et al., 2022), OlympiadBench (He et al., 2024), and MATH500 (Hendrycks et al., 2021c; Lightman et al., 2023), with details deferred to Appendix B.1. In the experiments, we report the avg@256 accuracy metric for the smaller AMC23 dataset, and avg@32 for other datasets following prior works (Zuo et al., 2025; Wang et al., 2025c). To reduce randomness in instance selection, we repeat the algorithms three times and report the average result. More details can be found in Appendix B.2.

**Baselines.** We compare the proposed CONST against various baselines, i.e., (1) NoFinetuning, which uses the original model for inference, (2) RandSelect, which annotates randomly selected instances for training, (3) active learning algorithms, including EntSampling (Settles, 1995), BADGE (Ash et al., 2020), and CEC (Safaei & Patel, 2025), and (4) reasoning-specific selection strategies, including SCF (Wang et al., 2022) and EWS (Beygelzimer et al., 2009). More details about the baseline methods can be found in Appendix B.3 and Appendix C.3.

**Implementation Details.** We adopt LLaMA-3.1-8B-Instruct (Grattafiori et al., 2024), DeepSeek-R1-Distill-Qwen-1.5B (Guo et al., 2025), Qwen2.5-Math-1.5B, and Qwen2.5-Math-7B (Yang et al., 2024) for RLVR training. For the calibration set, we use 1024 instances randomly selected from BigMath (ensuring no overlap with BigMath-sub), and to justify the robustness of CONST to the choice of the calibration set, MMLU (Hendrycks et al., 2021a;b) is used as an alternative. For procedural volatility, we set the number of stages $n_P$ to 20, and for outcome volatility, we set $n_O$ to 20. In conformal prediction, we set the error rate $\alpha$ to 0.1 and $\lambda$ to 0.02. For the budget of annotation, we report the results of 4 and 8 instances. For the clustering step, we use Sentence-BERT (Reimers & Gurevych, 2019) to obtain the embeddings of the input queries, and use the K-means algorithm to obtain the clusters. The number of clusters is set to $b$, which is the budget of annotation. For the RLVR optimization hyperparameter setup, we generally follow the training configuration of Wang et al. (2025c). By default, we set the maximum number of tokens to 8192 in training and 3072 in inference, the learning rate to $1 \times 10^{-6}$, the weight decay to 0.01, the hyperparameter of $\beta$ in GRPO (Eq. 2) to $1 \times 10^{-3}$, the batch size to 64, and 8 gradient updates for each rollout. We train the model for at most 500 iterations and evaluate the model every 20 iterations. During training, we duplicate the samples to occupy a single batch. We use the VERL framework (Sheng et al., 2024) for RLVR training and inference. For the computation hardware, we use 4 NVIDIA H800 for both training and inference.

Table 1: Performance comparison with various baselines and training on the full dataset under the `avg@32` (`avg@256` for AMC23) metric. We mark the best in **bold** and runner-ups with underline.

| Datasets | AMC23 | | MinervaMath | | OlympiadBench | | MATH500 | | AVG | |
|---|---|---|---|---|---|---|---|---|---|---|
| Budget | 4 | 8 | 4 | 8 | 4 | 8 | 4 | 8 | 4 | 8 |
| `LLaMA-3.1-8B-Instruct` | | | | | | | | | | |
| NoFinetuning | 18.03 | | 14.37 | | 12.28 | | 35.79 | | 20.12 | |
| RandSelect | 19.48 | 18.34 | 16.77 | 18.05 | 13.54 | 13.87 | 39.02 | 39.03 | 22.20 | 22.32 |
| EntSampling | 20.42 | 19.70 | 16.66 | 18.12 | 13.43 | 13.14 | 37.29 | 36.33 | 21.95 | 21.82 |
| BADGE | 19.34 | 20.02 | 20.25 | 19.28 | 13.27 | 13.81 | 41.72 | 42.68 | 23.95 | 23.65 |
| CEC | 20.25 | 21.24 | 16.66 | 18.50 | 13.05 | 15.02 | 38.03 | 42.97 | 21.79 | 24.43 |
| CONST (ours) | **20.62** | **24.27** | **21.68** | **24.19** | **16.83** | **17.61** | **43.46** | **47.17** | **25.65** | **28.31** |
| FullDataset | 24.30 | | 20.99 | | 18.23 | | 48.58 | | 28.03 | |
| `DeepSeek-R1-Distill-Qwen-1.5B` | | | | | | | | | | |
| NoFinetuning | 30.94 | | 14.17 | | 17.68 | | 50.03 | | 28.21 | |
| RandSelect | 40.96 | 54.55 | 19.28 | 23.31 | 25.02 | 33.68 | 64.29 | 72.96 | 37.39 | 46.13 |
| EntSampling | 54.91 | 54.88 | 22.23 | 22.55 | 32.68 | 32.98 | 71.79 | 71.97 | 45.40 | 45.60 |
| BADGE | 42.84 | 50.13 | 21.39 | 23.20 | 27.62 | 30.76 | 67.73 | 71.04 | 39.90 | 43.78 |
| CEC | 49.75 | 51.46 | 21.88 | 22.14 | 30.50 | 31.50 | 69.96 | 71.44 | 43.02 | 44.14 |
| CONST (ours) | **55.84** | **59.16** | **23.17** | **23.66** | **33.66** | **34.90** | **73.61** | **74.84** | **46.57** | **48.14** |
| FullDataset | 60.27 | | 24.55 | | 36.30 | | 75.49 | | 49.15 | |
| `Qwen2.5-Math-1.5B` | | | | | | | | | | |
| NoFinetuning | 31.74 | | 9.47 | | 21.72 | | 36.23 | | 24.79 | |
| RandSelect | 44.66 | 46.88 | 15.19 | 21.35 | 30.94 | 30.46 | 64.32 | 65.74 | 38.78 | 41.11 |
| EntSampling | 40.64 | 42.43 | 17.69 | 20.62 | 27.24 | 27.32 | 59.79 | 62.26 | 36.34 | 38.16 |
| BADGE | 44.61 | 46.73 | 19.91 | 20.97 | 28.87 | 29.54 | 63.69 | 64.88 | 39.27 | 40.53 |
| CEC | 42.42 | 45.32 | 13.67 | 21.24 | 28.00 | 30.88 | 59.11 | 68.87 | 35.80 | 41.58 |
| CONST (ours) | **47.19** | **48.42** | **20.01** | **24.54** | **32.05** | **32.75** | **67.68** | **69.58** | **41.73** | **43.82** |
| FullDataset | 49.13 | | 24.30 | | 33.33 | | 70.88 | | 44.41 | |
| `Qwen2.5-Math-7B` | | | | | | | | | | |
| NoFinetuning | 56.04 | | 33.90 | | 37.28 | | 81.03 | | 52.06 | |
| RandSelect | 57.05 | 57.48 | 35.26 | 35.91 | 38.29 | 38.77 | 81.58 | 81.90 | 53.05 | 53.52 |
| EntSampling | 56.95 | 57.87 | 35.36 | 35.37 | 38.46 | 38.63 | 81.88 | 81.89 | 53.16 | 53.44 |
| BADGE | 54.36 | 56.76 | 34.12 | 35.08 | 37.63 | 38.92 | 80.81 | 82.25 | 51.73 | 53.25 |
| CEC | 56.03 | 58.28 | 34.73 | 35.21 | 38.41 | 39.18 | 81.70 | 82.16 | 52.72 | 53.71 |
| CONST (ours) | **58.21** | **59.05** | **35.83** | **36.97** | **39.56** | **40.19** | **82.94** | **83.55** | **54.14** | **54.94** |
| FullDataset | 58.70 | | 36.66 | | 41.04 | | 83.61 | | 55.00 | |

## 4.2 PERFORMANCE COMPARISON

We compare the proposed `CONST` with various baselines across different LLM architectures on four mathematical reasoning datasets with varying budget sizes, and report the `avg@32` (`avg@256`) metric in Table 1. According to the results, we make several observations (Obs.) listed as follows: **Obs.❶ `CONST` significantly improves vanilla models across various scenarios, outperforming baselines.** For example, using only 8 critical instances, `CONST` significantly improves vanilla `LLaMA-3.1-8B-Instruct` by $40.71\%$, `DeepSeek-R1-Distill-Qwen-1.5B` by $70.65\%$, and `Qwen2.5-Math-1.5B` by $76.76\%$, surpassing competitive active learning baselines not designed for reinforcement learning on LLMs (*e.g.*, BADGE and CEC). **Obs.❷ Training on critical instances discovered by `CONST` achieves comparable results to training on the full dataset.** The results in Table 1 demonstrate that with less than $0.5\%$ of the samples selected by `CONST` *in an unsupervised manner*, we can achieve very similar performance on average: less than $1.09\%$ difference in terms of `avg@k` accuracy for budget 8. This confirms the value of these lottery-winning samples and our method that discovers them without access to ground truth answers.

## 4.3 ABLATION STUDIES

We then investigate how the different components or mechanisms in `CONST` affect the final performance. In particular, we design six variants of `CONST`, denoted as V1 to V6. V1 removes conformal prediction and randomly selects instances from each cluster. V2 skips clustering and chooses samples with the largest conformal prediction sets. V3 uses entropy in Eq. 8 instead of conformal prediction sets to select instances in each cluster. V4 explores alternative configurations in Eq. 11 by clustering instances into $b/2$ groups and selecting the top 2 items with the highest conformal

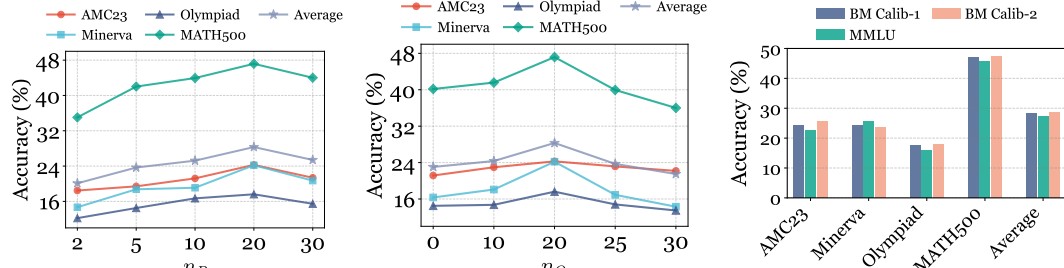

Figure 3: **Left:** performance under different numbers of stages ($n_P$). **Middle:** performance under different numbers of samples ($n_O$). **Right:** robustness to different choices of the calibration set.

prediction sets. V5 removes procedural volatility, and V6 removes outcome volatility. We perform experiments on `LLaMA-3.1-8B-Instruct` with the budget of 8, and the results in terms of `avg@32` (`avg@256`) are shown in Table 2, from which we have the following observations: **Obs.❸ All components or mechanisms in `CONST` contribute to the final performance.** As shown in the table, removing any of the proposed techniques consistently decreases accuracy across datasets, demonstrating the effectiveness of conformal prediction, procedural/outcome volatility, and clustering. **Obs.❹ Conformal prediction plays an important role.** The results show that replacing conformal prediction with alternatives such as random selection or entropy selection leads to severe performance degradation (*i.e.*, 5% drop in terms of absolute accuracy).

## 4.4 FURTHER ANALYSIS

**Performance under Different Hyperparameters.** We then show the model's performance (in terms of `avg@32`/`avg@256`) under different hyperparameters: the number of stages (*i.e.*, $n_P$) in procedural volatility (Eq. 4) and the number of samples (*i.e.*, $n_O$) in outcome volatility (Eq. 6). The experimental results on `LLaMA-3.1-8B-Instruct` are shown in Figure 3 (Left and Middle).

Table 2: Ablation study of `CONST` with the budget of 8.

| Variants | AMC23 | Minerva Math | Olympiad Bench | MATH 500 | AVG |
|---|---|---|---|---|---|
| V1 | 20.87 | 16.89 | 13.76 | 40.53 | 23.01 |
| V2 | 22.03 | 21.37 | 15.18 | 43.60 | 25.54 |
| V3 | 19.98 | 17.64 | 14.79 | 40.79 | 23.30 |
| V4 | 22.97 | 25.02 | 17.11 | 45.31 | 27.60 |
| V5 | 23.96 | 20.21 | 16.56 | 44.80 | 26.38 |
| V6 | 21.16 | 16.34 | 14.50 | 40.16 | 23.04 |
| **CONST** | 24.27 | 24.19 | 17.61 | 47.17 | 28.31 |

From the results, we observe that: **Obs.❺ Both $n_P$ and $n_O$ achieve best performance at** 20. The number of stages $n_P$ controls the granularity of procedural volatility: when the granularity is too coarse (small $n_P$s), it may be difficult to capture the twists and turns in the reasoning trajectories; when the granularity is small (large $n_P$s), it may frequently interrupt the logic fragments. On the other hand, $n_O$ controls the balance of procedural volatility and outcome volatility in the multiset $\mathcal{B}(X)$, which affects the scoring function and thus conformal prediction sets.

**Robustness to Different Choice of Calibration Sets.** `CONST` requires a calibration set, and therefore, we also investigate the method's robustness under different choices of the calibration sets. Specifically, we adopt three sets: (1) *BM Calib-1*, which is the original one; (2) *BM Calib-2*, which is also sampled from the BigMath dataset with no overlap with the training set; (3) *MMLU*, which contains mathematical questions from the MMLU dataset. The results on LLaMA are presented in Figure 3 (Right), and we observe that: **Obs.❻ The proposed method is robust to the choice of calibration sets.** Using a calibration set with identical distributions to the training set (*i.e.*, *BM Calib-1* and *BM Calib-2*) yields similar high accuracy on average. When using a calibration set with a different distribution (*i.e.*, *MMLU*), there is a slight decrease, but the difference is marginal on average, showing that `CONST` is robust to the different choices of the calibration set. The results also indicate that it is possible to use existing datasets that are already annotated (*e.g.*, MMLU) as the calibration set to avoid the need to annotate the calibration set.

## 5 RELATED WORKS

**Reinforcement Learning for LLM Reasoning.** LLMs and their transformer architecture have become a promising solution in many fields, including multimodal understanding (He et al., 2021;

Zhao et al., 2022; 2025b; Huang et al., 2025), formal languages (Zhao et al., 2025e), time-series understanding (Zhao et al., 2025d), and graph data (Zhao et al., 2025h). Reinforcement learning (Sutton et al., 1999; Havrilla et al., 2024a; Wen et al., 2024; Liu et al., 2025a) has significantly enhanced the reasoning capabilities of LLMs via rewards from verifiable answers (Shao et al., 2024; Mroueh, 2025; Wen et al., 2025) or reward models (Dong et al., 2024; Setlur et al., 2024; Qu et al., 2025). Early efforts (Sprueill et al., 2023; Deng et al., 2024; Wang et al., 2024) mainly focus on supervising the LLMs' reasoning process, often involving the value functions (Havrilla et al., 2024b; Zhai et al., 2025; Yuan et al., 2025; Zhang et al., 2025a). More recently, outcome supervision, with the reward obtained from verifiable ground truth answers, has received increasing attention (Shao et al., 2024; Liu et al., 2025b; Su et al., 2025; Liu et al., 2025c), due to its simplicity and the immunity from reward hacking (Gao et al., 2024; Fu et al., 2025; Miao et al., 2025).

**Conformal Prediction.** Conformal prediction (Vovk et al., 2005; Tibshirani et al., 2019; Angelopoulos et al., 2023; Straitouri et al., 2023; Kiyani et al., 2024a; Gibbs et al., 2025) is a model-agnostic and distribution-free solution of uncertainty quantification (Stracuzzi et al., 2017; Wang et al., 2019; Psaros et al., 2023), with solid mathematical foundations (Fontana et al., 2023; Angelopoulos et al., 2024). It generates prediction sets that contain the ground truth answer under a predefined error rate. While most prior efforts focus on conformal prediction with smaller classification or regression models (Correia et al., 2024; Jeary et al., 2024; Cresswell et al., 2024; Zhou et al., 2025), its adoption in natural language processing (Campos et al., 2024), and particularly LLMs (Cherian et al., 2024; Kiyani et al., 2024b; Su et al., 2024; Mohri & Hashimoto, 2024; Wang et al., 2025b; Chankaev & Ilyushin, 2025), has received increasing attention. Compared to these prior works, this paper uses conformal prediction to guide the selection of critical samples.

**Active Learning.** Active learning (Cohn et al., 1994; 1996; Baram et al., 2004; Castro et al., 2008; Ren et al., 2021) aims to optimize deep learning models with limited annotation efforts. It is particularly useful when the ground truth answers can only be obtained with relatively high costs (Konyushkova et al., 2017; Yuan et al., 2023; Xiao et al., 2023; Chen et al., 2024), and facilitates other tasks like test-time learning (Zhao et al., 2025a;g; Zuo et al., 2025). With the success of LLMs, efforts have been made in both LLM for active learning, which uses LLMs for active annotation (Margatina et al., 2023; Melo et al., 2024; Li et al., 2024; Kholodna et al., 2024; Ceravolo et al., 2024; Astorga et al., 2024; Xia et al., 2025), and active learning for LLMs, which adopts active learning for optimizing LLMs (Muldrew et al., 2024; Sun et al., 2024; Zhang et al., 2024a; Hübotter et al., 2024; Zhang et al., 2024b). In this paper, we explore active learning to optimize the reasoning capability of LLMs with a data-efficient and performance-competitive reasoning framework.

**Data Selection and Valuation.** Data selection and valuation (Das et al., 2020; Paul et al., 2021; Wang & Jia, 2023; Das et al., 2024; Ebiele et al., 2025) aim to find the most valuable data in the training set to save computational resources, which is also an important part of data-centric methods (Zhao et al., 2025f;c). For example, Paul et al. (Paul et al., 2021) use the loss function and its gradients to select important examples very early in training. Guo et al. (Guo et al., 2022) provide a comprehensive code library in addition to extensive evaluation for data subset selection and valuation. Das et al. (Das et al., 2024) propose CheckSelect, a flexible, accurate, robust, and efficient method for extracting the high-value subsets. Nevertheless, many works on data subset selection and valuation assume complete annotation of the training set by computing the loss function or its gradients (Paul et al., 2021; Wang & Jia, 2023; Das et al., 2024), and conducted primarily in the domain of vision (Das et al., 2020; 2021a; Paul et al., 2021). By comparison, this work proposes `CONST` that aims to find important training data (the critical instances) without the annotation, and only annotate the important instances to achieve annotation-efficient RLVR optimization of LLMs.

# 6 CONCLUSION

This paper investigates an important question: how can we identify the lottery-winning samples from the original dataset without access to answers, thereby enabling an annotation-minimal, data-efficient and performance-competitive alternative for optimizing the reasoning capability of LLMs. Through the design of our novel `CONST` framework, a probabilistic method grounded in the mathematical foundation of conformal prediction and incorporating complementary considerations of procedural and outcome volatility, we demonstrate that the unsupervised discovery of critical instances in full datasets can achieve comparable performance with significantly less annotation efforts.

## ACKNOLEDGEMENT

Ming Zhang and Yusheng Zhao are supported by grants from the National Natural Science Foundation of China (NSFC Grant Number 62276002). The authors are grateful to the anonymous reviewers for critically reading this article and for giving important suggestions to improve this article.

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

# A PROOF OF THEOREM 3.1

In this section, we prove our Theorem 3.1 in two steps. At first, in subsection A.1, we analyze the convergence of standard gradient descent, i.e., $\theta_{k+1} \triangleq \theta_k - \eta \nabla \hat{\mathcal{L}}_{\text{GRPO}}^{\mathcal{Q}'}(\theta)$, where $\eta$ denotes the step size and $\hat{\mathcal{L}}_{\text{GRPO}}^{\mathcal{Q}'}(\theta)$ represents the empirical GRPO loss function utilized by our `CONST` method. Subsequently, in Subsection A.2, we establish the generalization upper bound between the expected return $J(\theta) \triangleq \mathbb{E}_{\tau \sim \pi_\theta}[R(\tau)]$ and the empirical return, i.e., $-\hat{\mathcal{L}}_{\text{GRPO}}^{\mathcal{Q}}$.

## A.1 CONVERGENCE OF GRADIENT DESCENT

**Lemma A.1.** *Under Assumption 3.1-3.3, if the gradient $\nabla \hat{\mathcal{L}}_{GRPO}^{\mathcal{Q}}(\theta)$ is bounded, i.e., $\|\nabla \hat{\mathcal{L}}_{GRPO}^{\mathcal{Q}}(\theta)\|_2 \leq G$, then the Gradient Descent algorithm with a constant step-size $\frac{1}{L}$, that is,*

$$\theta_{k+1} \triangleq \theta_k - \frac{1}{L}\nabla \hat{\mathcal{L}}_{GRPO}^{\mathcal{Q}'}(\theta),$$

*has a linear convergence rate. We have*

$$\hat{\mathcal{L}}_{GRPO}^{\mathcal{Q}}(\theta_k) - \hat{\mathcal{L}}_{GRPO}^{\mathcal{Q}}(\theta_{GRPO}^*) \leq (1 - \frac{\mu}{L})^k \hat{\mathcal{L}}_{GRPO}^{Q}(\theta_0) + \frac{2G}{\mu}\epsilon + \frac{\epsilon^2}{2\mu},$$

*where $\theta_{GRPO}^* \triangleq \arg\min_\theta \hat{\mathcal{L}}_{GRPO}^{Q}(\theta)$.*

*Proof.* According to the $L$-smoothness (Lan, 2020), we have that

$$\hat{\mathcal{L}}_{\text{GRPO}}^{\mathcal{Q}}(\theta_{k+1}) \leq \hat{\mathcal{L}}_{\text{GRPO}}^{\mathcal{Q}}(\theta_k) + \langle \nabla \hat{\mathcal{L}}_{\text{GRPO}}^{\mathcal{Q}}(\theta_k), \theta_{k+1} - \theta_k \rangle + \frac{L}{2}\|\theta_{k+1} - \theta_k\|_2^2$$

$$= \hat{\mathcal{L}}_{\text{GRPO}}^{\mathcal{Q}}(\theta_k) - \frac{1}{L}\langle \nabla \hat{\mathcal{L}}_{\text{GRPO}}^{\mathcal{Q}}(\theta_k), \nabla \hat{\mathcal{L}}_{\text{GRPO}}^{\mathcal{Q}'}(\theta) \rangle + \frac{1}{2L}\|\nabla \hat{\mathcal{L}}_{\text{GRPO}}^{\mathcal{Q}'}(\theta_k)\|_2^2,$$

where the final equality follows from $\theta_{k+1} \triangleq \theta_k - \frac{1}{L}\nabla \hat{\mathcal{L}}_{\text{GRPO}}^{\mathcal{Q}'}(\theta)$.

Next, we show that

$$\langle \nabla \hat{\mathcal{L}}_{\text{GRPO}}^{\mathcal{Q}}(\theta_k), \nabla \hat{\mathcal{L}}_{\text{GRPO}}^{\mathcal{Q}'}(\theta) \rangle = \langle \nabla \hat{\mathcal{L}}_{\text{GRPO}}^{\mathcal{Q}}(\theta_k), \nabla \hat{\mathcal{L}}_{\text{GRPO}}^{\mathcal{Q}}(\theta) \rangle + \langle \nabla \hat{\mathcal{L}}_{\text{GRPO}}^{\mathcal{Q}}(\theta_k), \nabla \hat{\mathcal{L}}_{\text{GRPO}}^{\mathcal{Q}'}(\theta) - \nabla \hat{\mathcal{L}}_{\text{GRPO}}^{\mathcal{Q}}(\theta) \rangle$$

$$\geq \|\nabla \hat{\mathcal{L}}_{\text{GRPO}}^{\mathcal{Q}}(\theta_k)\|_2^2 - \|\nabla \hat{\mathcal{L}}_{\text{GRPO}}^{\mathcal{Q}}(\theta_k)\|_2 \|\nabla \hat{\mathcal{L}}_{\text{GRPO}}^{\mathcal{Q}'}(\theta) - \nabla \hat{\mathcal{L}}_{\text{GRPO}}^{\mathcal{Q}}(\theta)\|_2 \geq \|\nabla \hat{\mathcal{L}}_{\text{GRPO}}^{\mathcal{Q}}(\theta_k)\|_2^2 - \epsilon G,$$

where the final inequality from the Assumption 3.1 and boundedness. Moreover,

$$\|\nabla \hat{\mathcal{L}}_{\text{GRPO}}^{\mathcal{Q}'}(\theta_k)\|_2^2$$

$$= \|\nabla \hat{\mathcal{L}}_{\text{GRPO}}^{\mathcal{Q}}(\theta_k)\|_2^2 + 2\langle \nabla \hat{\mathcal{L}}_{\text{GRPO}}^{\mathcal{Q}}(\theta_k), \nabla \hat{\mathcal{L}}_{\text{GRPO}}^{\mathcal{Q}'}(\theta_k) - \nabla \hat{\mathcal{L}}_{\text{GRPO}}^{\mathcal{Q}}(\theta) \rangle + \|\nabla \hat{\mathcal{L}}_{\text{GRPO}}^{\mathcal{Q}'}(\theta_k) - \nabla \hat{\mathcal{L}}_{\text{GRPO}}^{\mathcal{Q}}(\theta_k)\|_2^2$$

$$\leq \|\nabla \hat{\mathcal{L}}_{\text{GRPO}}^{\mathcal{Q}}(\theta_k)\|_2^2 + 2G\epsilon + \epsilon^2,$$

where the final inequality from the Assumption 3.1 and boundedness.

As a result, we have that

$$\hat{\mathcal{L}}_{\text{GRPO}}^{\mathcal{Q}}(\theta_{k+1}) \leq \hat{\mathcal{L}}_{\text{GRPO}}^{\mathcal{Q}}(\theta_k) - \frac{1}{2L}\|\nabla \hat{\mathcal{L}}_{\text{GRPO}}^{\mathcal{Q}}(\theta_k)\|_2^2 + \frac{2G}{L}\epsilon + \frac{\epsilon^2}{2L}$$

$$\leq \hat{\mathcal{L}}_{\text{GRPO}}^{\mathcal{Q}}(\theta_k) - \frac{\mu}{L}\big(\hat{\mathcal{L}}_{\text{GRPO}}^{Q}(\theta_k) - \hat{\mathcal{L}}_{\text{GRPO}}^{Q}(\theta_{\text{GRPO}}^*)\big) + \frac{2G}{L}\epsilon + \frac{\epsilon^2}{2L},$$

where the final inequality follows from Assumption 3.3.

Finally, we have

$$\hat{\mathcal{L}}_{\text{GRPO}}^{\mathcal{Q}}(\theta_{k+1}) - \hat{\mathcal{L}}_{\text{GRPO}}^{\mathcal{Q}}(\theta_{\text{GRPO}}^*)$$

$$\leq (1 - \frac{\mu}{L})\big(\hat{\mathcal{L}}_{\text{GRPO}}^{Q}(\theta_k) - \hat{\mathcal{L}}_{\text{GRPO}}^{Q}(\theta_{\text{GRPO}}^*)\big) + \frac{2G}{L}\epsilon + \frac{\epsilon^2}{2L}$$

$$\leq \cdots$$

$$\leq (1 - \frac{\mu}{L})^{k+1}\big(\hat{\mathcal{L}}_{\text{GRPO}}^{Q}(\theta_0) - \hat{\mathcal{L}}_{\text{GRPO}}^{Q}(\theta_{\text{GRPO}}^*)\big) + (\frac{2G}{L}\epsilon + \frac{\epsilon^2}{2L})\sum_{j=0}^{k}(1 - \frac{\mu}{L})^j$$

$$\leq (1 - \frac{\mu}{L})^{k+1}\big(\hat{\mathcal{L}}_{\text{GRPO}}^{Q}(\theta_0) - \hat{\mathcal{L}}_{\text{GRPO}}^{Q}(\theta_{\text{GRPO}}^*)\big) + \frac{2G}{\mu}\epsilon + \frac{\epsilon^2}{2\mu},$$

where the final inequality follows from $\sum_{j=0}^{k}(1-\frac{\mu}{L})^j \leq \frac{L}{\mu}$.

$\square$

### A.2 GENERALIZATON OF GRPO

**Lemma A.2.** *If the underlying MDP $\mathcal{M} \triangleq (\mathcal{S}, \mathcal{A}, P, r, \gamma)$ is ergodic with mixed time $t_{mix}$, then we can show that, for any $\delta \in (0,1)$, with probability $1 - \delta$, the following inequality holds, that is,*

$$\sup_{\theta \in \Theta} |\hat{\mathcal{L}}_{GRPO}^{\mathcal{Q}}(\theta) - \mathcal{L}_{GRPO}(\theta)| \leq 2\mathcal{R}(\mathcal{F}_{GR}) + \mathcal{O}(\sqrt{\frac{t_{mix}\sigma_R^2(1-\frac{1}{n})\ln(\frac{1}{\delta})}{Nn}} + \frac{\ln(\frac{1}{\delta})}{Nn(1-\gamma)}),$$

*where $\Theta$ denotes the parameter space, $\mathcal{R}(\mathcal{F}_{GR})$ is the Rademacher complexity of the group-relative loss function space $\mathcal{F}_{GR}$, $N$ denotes the size of full training set $\mathcal{Q}$, $n$ is size of outputs for each question and $\sigma_R^2$ is an upper bound of variance of the return $\{r_i\}_{i=1}^n$, i.e., $Var_{\pi_\theta}(r_i) \leq \sigma_R^2, \forall \theta \in \Theta$.*

*Proof.* At first, we introduce a theorem from Tolstikhin & Seldin (2013), that is,

**Theorem A.1** (Lemma 1 in Tolstikhin & Seldin (2013)). *For any function $f_n : \mathcal{H} \times (\mathcal{X} \times \mathcal{Y})^n \to \mathbb{R}$ and for any distribution $P_1$ over $\mathcal{H}$, such that $P_1$ is independent of dataset $S \triangleq [(x_1,y_1),\ldots,(x_n,y_n)]$, with probability greater than $1 - \delta$ over a random draw of S, for all distributions $P_2$ over $\mathcal{H}$ simultaneously:*

$$\mathbb{E}_{h \sim P_2}[f_n(h,S)] \leq KL(P_2|P_1) + \ln(\frac{1}{\delta}) + \ln\left(\mathbb{E}_{h \sim P_1}\left[\mathbb{E}_{S' \sim \mathcal{D}^n}\left[e^{f_n(h,S')}\right]\right]\right), \quad (12)$$

*where $S' \sim \mathcal{D}^n$ represent a $n$-size independent dataset $S'$ drawn from data space $\mathcal{D}$.*

Then, we set $f_N(\theta, \mathcal{Q}) = |\hat{\mathcal{L}}_{GRPO}^{\mathcal{Q}}(\theta) - \mathcal{L}_{GRPO}(\theta)|$ and investigate the expectation $\mathbb{E}_{\mathcal{Q}}\left[e^{\lambda(f_N(\theta,\mathcal{Q})-\mathbb{E}_{\mathcal{Q}}[f_N(\theta,\mathcal{Q})])}\right]$ for any fixed $\theta \in \Theta$ and $\lambda > 0$. From the classic Bernstein-type self-bounding inequality(e.g. Theorem 2.1. in Fan & Shao (2025)), we can have that

$$\mathbb{E}_{\mathcal{Q}}\left[e^{\lambda(f_N(\theta,\mathcal{Q})-\mathbb{E}_{\mathcal{Q}}[f_N(\theta,\mathcal{Q})])}\right] \leq \exp(\frac{\lambda^2 V}{2(1 - \lambda\frac{nN\Delta}{3})}),$$

where $V = \sum_{j=1}^N \mathbb{E}\left[\left(\hat{\mathcal{L}}_{GRPO}^{\mathcal{Q}}(\theta) - \hat{\mathcal{L}}_{GRPO}^{\mathcal{Q}\backslash X_j}(\theta)\right)^2\right]$, $\left|\hat{\mathcal{L}}_{GRPO}^{\mathcal{Q}}(\theta) - \hat{\mathcal{L}}_{GRPO}^{\mathcal{Q}\backslash X_j}(\theta)\right| \leq \Delta$ deterministically for any $j \in [N]$ and $\hat{\mathcal{L}}_{GRPO}^{\mathcal{Q}\backslash X_j}(\theta)$ represent the leave-one-question-out loss.

In standard MDP, we usually use the discount return such that we could infer that $\Delta = \mathcal{O}(\frac{(1-\gamma)^{-1}}{Nn})$ where $\gamma$ is the discount parameter (Puterman, 1990) . Moreover, from the variance conversion for mixing ergodic MDP (Levin & Peres, 2017), we also can show that $V \leq \frac{t_{min}\sigma_R^2(1-\frac{1}{n})}{Nn}$.

From Eq.12, we can have that

$$\mathbb{E}_{\theta \sim P_2}[\lambda(f_N(\theta,\mathcal{Q}) - \mathbb{E}_{\mathcal{Q}}[f_N(\theta,\mathcal{Q})])]$$
$$\leq KL(P_2|P_1) + \ln(\frac{1}{\delta}) + \ln\left(\mathbb{E}_{\theta \sim P_1}\left[\mathbb{E}_{\mathcal{Q}}\left[e^{\lambda(f_N(\theta,\mathcal{Q})-\mathbb{E}_{\mathcal{Q}}[f_N(\theta,\mathcal{Q})])}\right]\right]\right)$$
$$\leq KL(P_2|P_1) + \ln(\frac{1}{\delta}) + \frac{\lambda^2 t_{mix}\sigma_R^2(1-\frac{1}{n})}{2Nn(1 - \lambda(1-\gamma)^{-1}/3)}.$$

Let $P_1 = P_2$, we thus can show that

$$\mathbb{E}_{\theta \sim P_2}[|\hat{\mathcal{L}}_{GRPO}^{\mathcal{Q}}(\theta) - \mathcal{L}_{GRPO}(\theta)|]$$
$$\leq \mathbb{E}_{\theta \sim P_2}\left[\mathbb{E}_{\mathcal{Q}}[|\hat{\mathcal{L}}_{GRPO}^{\mathcal{Q}}(\theta) - \mathcal{L}_{GRPO}(\theta)|]\right] + \frac{\ln(\frac{1}{\delta})}{\lambda} + \frac{\lambda t_{mix}\sigma_R^2(1-\frac{1}{n})}{2Nn(1 - \lambda(1-\gamma)^{-1}/3)}$$
$$\leq \mathbb{E}_{\mathcal{Q}}[\sup_{\theta \in \Theta}|\hat{\mathcal{L}}_{GRPO}^{\mathcal{Q}}(\theta) - \mathcal{L}_{GRPO}(\theta)|] + \frac{\ln(\frac{1}{\delta})}{\lambda} + \frac{\lambda t_{mix}\sigma_R^2(1-\frac{1}{n})}{2Nn(1 - \lambda(1-\gamma)^{-1}/3)},$$

where the final inequality follows from $|\hat{\mathcal{L}}_{GRPO}^{\mathcal{Q}}(\theta) - \mathcal{L}_{GRPO}(\theta)| \leq \sup_{\theta \in \Theta}|\hat{\mathcal{L}}_{GRPO}^{\mathcal{Q}}(\theta) - \mathcal{L}_{GRPO}(\theta)|$.

Like the structure of the proof of Theorem 3 of Tolstikhin & Seldin (2013), we can investigate the function $g(\lambda) \triangleq \frac{\ln(\frac{1}{\delta})}{\lambda} + \frac{\lambda t_{mix}\sigma_R^2(1-\frac{1}{n})}{2Nn(1-\lambda(1-\gamma)^{-1}/3)}$ where $\lambda \in (0, \frac{3}{1-\lambda})$. The minimum of $g(\lambda)$ often occurs at $\lambda^* = \sqrt{\frac{2Nn\log(\frac{1}{\delta})}{t_{min}\sigma_R^2(1-\frac{1}{n})(1+\beta^*)}}$ and $\beta^* = \frac{(1-\gamma)^{-1}\lambda^*}{3}$ such that $g(\lambda^*) = \mathcal{O}(\sqrt{\frac{t_{mix}\sigma_R^2(1-\frac{1}{n})\ln(\frac{1}{\delta})}{Nn}} + \frac{\ln(\frac{1}{\delta})}{Nn(1-\gamma)})$. As a result, we have that

$$\mathbb{E}_{\theta \sim P_2}[|\hat{\mathcal{L}}_{\text{GRPO}}^{\mathcal{Q}}(\theta) - \mathcal{L}_{\text{GRPO}}(\theta)|]$$

$$\leq \mathbb{E}_{\mathcal{Q}}[\sup_{\theta \in \Theta} |\hat{\mathcal{L}}_{\text{GRPO}}^{\mathcal{Q}}(\theta) - \mathcal{L}_{\text{GRPO}}(\theta)|] + \mathcal{O}(\sqrt{\frac{t_{mix}\sigma_R^2(1-\frac{1}{n})\ln(\frac{1}{\delta})}{Nn}} + \frac{\ln(\frac{1}{\delta})}{Nn(1-\gamma)}).$$

Furthermore, from the classical symmetrization lemma in statistical learning theory (Shalev-Shwartz & Ben-David, 2014; Mitzenmacher & Upfal, 2017), we have that $\mathbb{E}_{\mathcal{Q}}[\sup_{\theta \in \Theta} |\hat{\mathcal{L}}_{\text{GRPO}}^{\mathcal{Q}}(\theta) - \mathcal{L}_{\text{GRPO}}(\theta)|] \leq 2\mathcal{R}(\mathcal{F}_{GR})$ where $\mathcal{R}(\mathcal{F}_{GR})$ is the Rademacher complexity of the group-relative loss function space $\mathcal{F}_{GR}$. Therefore, we have that

$$\mathbb{E}_{\theta \sim P_2}[|\hat{\mathcal{L}}_{\text{GRPO}}^{\mathcal{Q}}(\theta) - \mathcal{L}_{\text{GRPO}}(\theta)|] \leq 2\mathcal{R}(\mathcal{F}_{GR}) + \mathcal{O}(\sqrt{\frac{t_{mix}\sigma_R^2(1-\frac{1}{n})\ln(\frac{1}{\delta})}{Nn}} + \frac{\ln(\frac{1}{\delta})}{Nn(1-\gamma)}).$$

Finally, due to the randomness of $P_2$ over $\Theta$, we get the result of lemma A.2. $\qquad \square$

With the results of Lemma A.1 and Lemma A.2, we also prove Theorem 3.1, that is,

$$\mathcal{L}_{\text{GRPO}}(\theta_k) - \mathcal{L}_{\text{GRPO}}(\theta^*)$$
$$= \mathcal{L}_{\text{GRPO}}(\theta_k) - \hat{\mathcal{L}}_{\text{GRPO}}^{\mathcal{Q}}(\theta_k) + \hat{\mathcal{L}}_{\text{GRPO}}^{\mathcal{Q}}(\theta_k) - \hat{\mathcal{L}}_{\text{GRPO}}^{\mathcal{Q}}(\theta^*) + \hat{\mathcal{L}}_{\text{GRPO}}^{\mathcal{Q}}(\theta^*) - \mathcal{L}_{\text{GRPO}}(\theta^*)$$
$$\leq 2\sup_{\theta \in \Theta} |\hat{\mathcal{L}}_{\text{GRPO}}^{\mathcal{Q}}(\theta) - \mathcal{L}_{\text{GRPO}}(\theta)| + \hat{\mathcal{L}}_{\text{GRPO}}^{\mathcal{Q}}(\theta_k) - \hat{\mathcal{L}}_{\text{GRPO}}^{\mathcal{Q}}(\theta_{\text{GRPO}}^*)$$

$$\leq 4\mathcal{R}(\mathcal{F}_{GR}) + (1 - \frac{\mu}{L})^k \hat{\mathcal{L}}_{\text{GRPO}}^{Q}(\theta_0) + \mathcal{O}(\sqrt{\frac{t_{mix}\sigma_R^2(1-\frac{1}{n})\ln(\frac{1}{\delta})}{Nn}} + \frac{\ln(\frac{1}{\delta})}{Nn(1-\gamma)} + \frac{2G}{\mu}\epsilon + \frac{\epsilon^2}{2\mu})$$

where the first inequality comes from $\theta_{\text{GRPO}}^* \triangleq \arg\min_\theta \hat{\mathcal{L}}_{\text{GRPO}}^{Q}(\theta)$ and $\theta^* \triangleq \arg\min_\theta \mathcal{L}_{\text{GRPO}}(\theta)$.

# B DETAILS OF THE EXPERIMENTAL SETUP

## B.1 MORE DETAILS ABOUT THE DATASET

Our experimental evaluation is conducted on several widely recognized mathematical reasoning benchmarks. For model training, we utilize a subset of the BigMath dataset, while for the calibration of our scoring function, we use instances from both BigMath and MMLU. For the test phase, our evaluation spans four distinct datasets to ensure a comprehensive assessment of performance. Below, we provide a detailed description of each dataset.

- **BigMath-sub.** For the training phase of our experiments, we use BigMath-sub, which is a randomly selected subset containing 2048 instances from the large-scale BigMath dataset (Albalak et al., 2025). BigMath is a high-quality dataset specifically curated for reinforcement learning in large language models on mathematical tasks.
- **MMLU.** The Massive Multitask Language Understanding (MMLU) dataset (Hendrycks et al., 2021a;b) is a comprehensive benchmark to evaluate the reasoning ability of LLMs. In our experiments, it serves as an alternative calibration set to justify the robustness of our method against the choice of calibration data. To maintain relevance to our mathematical reasoning task, we utilize a specific subset of MMLU, comprising five math-related subjects.
- **AMC23.** The American Mathematics Competitions (AMC) dataset (problems & solutions, 2023) is a collection of challenging problems from the official mathematics competitions for students in the US. These problems require both the knowledge of mathematical concepts and excellent problem-solving ability.
- **MinervaMath.** This dataset (Lewkowycz et al., 2022) is a benchmark focused on quantitative reasoning, containing problems that require step-by-step logical inference. The problems are sourced from various STEM disciplines and are designed to test the model's ability to perform complex, multi-step calculations and reasoning.
- **OlympiadBench.** This is a highly challenging benchmark consisting of problems from international science and mathematics Olympiads (He et al., 2024). The dataset is designed to push the limits of LLM reasoning capabilities, as the problems often require creative and non-standard approaches to solve.
- **MATH500.** The MATH dataset (Hendrycks et al., 2021c; Lightman et al., 2023) is a widely adopted benchmark for mathematical problem-solving, composed of 12,500 problems from high school math competitions. The problems are categorized by difficulty and subject, covering topics such as algebra, geometry, number theory, and more. We use a 500-instance subset for our evaluation.

## B.2 MORE DETAILS ABOUT THE EVALUATION METRICS

In the experiments, we use the `avg@k` accuracy metric for evaluating the performance. This metric is formally defined as follows. Given an input $X$ and the ground truth answer $Y$, the model predicts a total of $k$ answers, *i.e.*, $\widehat{Y}_1, \widehat{Y}_2, ..., \widehat{Y}_k$, and the metric can be computed as:

$$\texttt{avg@k} = \frac{\sum_{i=1}^{k} \mathbb{1}[\widehat{Y}_i = Y]}{k}, \tag{13}$$

where $\mathbb{1}$ is the indicator function. In the evaluation, we consider the size of the datasets when deciding $k$: for the smaller dataset of AMC23, we set $k$ to 256, whereas for larger datasets of MinervaMath, OlympiadBench, and MATH500, we set $k$ to 32.

## B.3 MORE DETAILS ABOUT THE BASELINE METHODS

We compare the proposed `CONST` against various baselines. These methods are grouped into three categories: a non-finetuning baseline, a random selection baseline, and several active learning algorithms. A detailed description of each baseline is provided below.

- **NoFinetuning.** This baseline directly uses the original large language model for inference without any fine-tuning. It serves as a fundamental benchmark to evaluate the performance improvement brought by different instance selection and training strategies.

- **RandSelect.** This is a simple yet crucial baseline where a subset of instances is randomly selected from the entire training set for annotation and subsequent model optimization. This method helps to gauge the effectiveness of more sophisticated active learning strategies against a naive selection approach.

- **EntSampling.** As a classic uncertainty-based active learning method, Entropy Sampling (Settles, 1995) selects instances for which the model has the highest predictive entropy. Our implementation adapts this principle for generative LLMs. For each instance, we first generate a multiset of potential answers by sampling the model's output $n_o$ times. The entropy is then calculated based on the frequency distribution of the unique answers within this multiset. This procedure directly mirrors the Outcome Volatility component of our proposed `CONST` method, ensuring a fair comparison.

- **BADGE.** Deep Batch Active learning by Diverse, Uncertain Gradient Lower Bounds (BADGE) (Ash et al., 2020) is a state-of-the-art active learning strategy that selects a batch of samples that are both uncertain and diverse. It computes a hypothetical gradient embedding for each unlabeled sample with respect to the parameters of the final layer. The magnitude of this gradient represents the model's uncertainty. To ensure diversity, it then uses k-MEANS++ seeding on these gradient embeddings to select a batch of points that are far from each other in the gradient space.

- **CEC.** Calibrated Entropy-weighted Clustering (CEC) (Safaei & Patel, 2025) is an active learning framework designed for vision-language models that selects informative samples by leveraging both self-uncertainty and neighbor-aware uncertainty. The method first calibrates the predictive entropy to counteract the known miscalibration issue in large pre-trained models. It then combines this calibrated self-uncertainty with the uncertainty of neighboring samples to obtain a more reliable score. Finally, CEC employs an uncertainty-weighted clustering approach to ensure diverse sample selection from various regions of the feature space. For our LLM-based tasks, we apply its core mechanisms directly to the language model's final-layer embeddings.

## C  CASE STUDY

In this section, we present three case studies to qualitatively demonstrate the effectiveness of our proposed `CONST` method. For each case, we show the question, the response from the original LLM (Before Training), and the response from the LLM fine-tuned on critical instances selected by `CONST` (After Training). An analysis is provided to highlight the specific improvements in the model's reasoning capabilities.

### C.1  EXAMPLE

---

**Example 1: Question**

*Given that the sum of the distances from any point inside a regular tetrahedron with edge length $a$ to its four faces is a constant value, find this value.*

---

**Example 1: Before Training (Original Model)**

To find the constant value of the sum of the distances from any point inside a regular tetrahedron to its four faces, we can use the following steps:

1. Consider a regular tetrahedron with edge length $a$.

2. Let $P$ be any point inside the tetrahedron. We want to find the sum of the distances from $P$ to the four faces.

3. The sum of the distances can be expressed as $d_1 + d_2 + d_3 + d_4$.

4. We can use the fact that the sum of the distances from a point inside a tetrahedron to its four faces is equal to the height of the tetrahedron multiplied by 3.

5. The height of a regular tetrahedron with edge length $a$ can be found using the formula $h = \frac{a\sqrt{6}}{3}$.

6. Therefore, the sum of the distances is $3h = 3 \cdot \frac{a\sqrt{6}}{3} = a\sqrt{6}$.

The final answer is: $\boxed{a\sqrt{6}}$.

---

**Example 1: After Training (Fine-tuned Model)**

**Step 1: Understanding the Problem.** We are given a regular tetrahedron with edge length $a$. The problem asks for the constant value that is the sum of the distances from any point inside the tetrahedron to its four faces.

**Step 2: Identifying Key Geometric Properties.** A regular tetrahedron has four equilateral triangular faces. A key result (a generalization of Viviani's theorem) states that for any interior point, the sum of the distances to the four faces is a constant, equal to the tetrahedron's height.

**Step 3: Considering the Centroid.** The distance $d$ from the centroid to each face can be calculated using the formula $d = \frac{a\sqrt{6}}{12}$.

**Step 4: Calculating the Height and Summing the Distances.** Since there are four faces, the sum of the distances from the centroid (or any point inside the tetrahedron) to the four faces is $4d = 4 \times \frac{a\sqrt{6}}{12} = \frac{a\sqrt{6}}{3}$.

**Step 5: Conclusion.** Therefore, the constant value that is the sum of the distances from any point inside the regular tetrahedron to its four faces is $\frac{a\sqrt{6}}{3}$.

The final answer is: $\boxed{\frac{\sqrt{6}}{3}a}$

---

**Analysis:** The original model makes a fundamental error in geometric reasoning, incorrectly stating that the sum of distances is three times the height (Step 4). This conceptual mistake leads directly to a wrong final answer. After fine-tuning, the model demonstrates a complete conceptual and procedural correction. It correctly identifies the key geometric principle: the sum of the distances is equal to the height of the tetrahedron (Step 2). Furthermore, it executes a sound, step-by-step calculation

to correctly derive the height's value as $\frac{a\sqrt{6}}{3}$ and provides the correct final answer. This successful transformation from a flawed understanding to a fully correct solution showcases the effectiveness of the training sample selected by CONST.

---

**Example 2: Question**

*A school selects 50 students from all 500 senior students for a study situation questionnaire survey. The 500 students are numbered from 1 to 500. The interval number $k = \frac{500}{50} = 10$, which means one student is selected out of every 10 students. A number is randomly selected from 1 to 10, and if the number 6 is selected, the numbers that should be taken from 125 to 140 are ______.*

---

**Example 2: Before Training (Original Model)**

To find the numbers that should be taken from 125 to 140, we need to determine which numbers in this range correspond to the interval of 10.
Since the interval number k = 10, we can divide the range 125 to 140 into intervals of 10.
The problem states that if the number 6 is selected, the numbers that should be taken from 125 to 140 are ______. This means that we need to select every 10th number starting from 125.
So, the numbers that should be taken from 125 to 140 are: 125, 135.
The final answer is: $\boxed{125, 135}$.

---

**Example 2: After Training (Fine-tuned Model)**

**Step 1: Understand the selection process.** The school is using systematic sampling. One student is chosen from every 10 students.

**Step 2: Determine the selection pattern.** A random start from 1 to 10 was chosen, and it was 6. This means the selected students will be those numbered 6, 16, 26, 36, and so on. The general form for a selected student's number is $10n + 6$ for some integer $n \geq 0$.

**Step 3: Identify the relevant range of students.** We are interested in the numbers that should be taken from 125 to 140.

**Step 4: Apply the selection pattern to the relevant range.** We need to find numbers of the form $10n + 6$ that fall between 125 and 140.

- If $n = 11$, the number is $10(11) + 6 = 116$ (too low).
- If $n = 12$, the number is $10(12) + 6 = 126$ (in the range).
- If $n = 13$, the number is $10(13) + 6 = 136$ (in the range).
- If $n = 14$, the number is $10(14) + 6 = 146$ (too high).

**Step 5: List the selected numbers in the range 125 to 140.** The selected numbers are 126 and 136.

The final answer is: $\boxed{126 \text{ and } 136}$

---

**Analysis:** The original model completely misunderstands the concept of systematic sampling; it ignores the provided random start (6) and incorrectly assumes the selection begins at the start of the local range (125), leading to a wrong answer. In sharp contrast, the model after fine-tuning demonstrates a perfect understanding of the procedure. It correctly identifies the general form of the selected numbers ($10n + 6$) and flawlessly applies this rule to the specified range to find the two correct numbers, 126 and 136. This case illustrates a clear and complete correction from flawed logic to precise procedural execution, with both the reasoning and the final answer being correct.

---

**Example 3: Question**

*Given that the sequence $\{b_n\}$ is an arithmetic sequence with the first term as -34 and a common difference of 1, and the sequence $\{a_n\}$ satisfies $a_{n+1} - a_n = 2^n$ ($n \in \mathbb{N}^*$), and $a_1 = b_{37}$, then the maximum value of the sequence $\left\{\frac{b_n}{a_n}\right\}$ is ______.*

---

---

**Example 3: Before Training (Original Model)**

Given that $\{b_n\}$ is an arithmetic sequence with the first term as -34 and a common difference of 1, we can write the formula for $b_n$ as: $b_n = -34 + (n-1) \cdot 1 = -33 + n$.

Given that $a_{n+1} - a_n = 2^n$, we can write the formula for $a_n$ as: $a_n = a_1 + \sum_{k=1}^{n-1} 2^k = a_1 + 2^n - 2$.
Since $a_1 = b_{37}$, we can substitute $n = 37$ into the formula for $b_n$ to find $a_1$: $a_1 = b_{37} = -33 + 37 = 4$. So, $a_n = 4 + 2^n - 2 = 2 + 2^n$.

Now, we can find the value of $\frac{b_n}{a_n} = \frac{-33+n}{2+2^n}$.

To find the maximum value of the sequence, we can try different values of $n$: $n = 5, \frac{-28}{34}$; $n = 6, \frac{-27}{66}$. We can see that the value of $n$ that makes the denominator close to the numerator is not an integer. We can try the value of $n = 3$: $\frac{-30}{10} = -3$. ... [The model continues to plug in values without a clear strategy and fails to find the maximum.]

---

**Example 3: After Training (Fine-tuned Model)**

**Step 1: Determine the general term of the arithmetic sequence $\{b_n\}$.** Given $b_1 = -34$ and common difference $d = 1$, the general term is: $b_n = b_1 + (n-1)d = -34 + (n-1)(1) = n - 35$.

**Step 2: Determine the general term of the sequence $\{a_n\}$.** We are given the recurrence $a_{n+1} - a_n = 2^n$. The general term $a_n$ can be found by telescoping sum: $a_n = a_1 + \sum_{k=1}^{n-1}(a_{k+1} - a_k) = a_1 + \sum_{k=1}^{n-1} 2^k$. This is a geometric series sum: $\sum_{k=1}^{n-1} 2^k = 2(2^{n-1} - 1) = 2^n - 2$. So, $a_n = a_1 + 2^n - 2$.

**Step 3: Find the value of $a_1$.** We are given $a_1 = b_{37}$. Using our formula for $b_n$: $a_1 = 37 - 35 = 2$.

**Step 4: Finalize the formula for $a_n$.** Substituting $a_1 = 2$ into the expression from Step 2: $a_n = (2) + 2^n - 2 = 2^n$.

**Step 5: Find the ratio $\frac{b_n}{a_n}$.** The sequence we want to maximize is $\left\{ \frac{n-35}{2^n} \right\}$.

**Step 6: Determine the maximum value.** Let $f(n) = \frac{n-35}{2^n}$. The denominator grows much faster than the numerator. For $n < 35$, the value is negative. For $n = 35$, the value is 0. For $n > 35$, the value is positive and will eventually decrease towards 0. The maximum value will occur for a small $n \geq 36$. Let's check $n = 36$: $f(36) = \frac{36-35}{2^{36}} = \frac{1}{2^{36}}$. Let's check $n = 37$: $f(37) = \frac{37-35}{2^{37}} = \frac{2}{2^{37}} = \frac{1}{2^{36}}$. Let's check $n = 38$: $f(38) = \frac{38-35}{2^{38}} = \frac{3}{2^{38}}$. Since $\frac{3}{2^{38}} < \frac{4}{2^{38}} = \frac{1}{2^{36}}$, the value decreases after $n = 37$. The maximum value is $\frac{1}{2^{36}}$.

The final answer is: $\boxed{\dfrac{1}{2^{36}}}$

---

**Analysis:** This multi-step problem requires correctly deriving formulas for two sequences before optimizing their ratio. The original model makes errors in deriving both $b_n$ (off-by-one) and $a_n$, making the rest of the problem unsolvable. After training, the model flawlessly executes the setup phase: it correctly derives $b_n = n - 35$ and $a_n = 2^n$. This mastery of the prerequisite steps is a significant improvement. It then correctly analyzes the behavior of the resulting function $\frac{n-35}{2^n}$ and, by testing points near the transition from negative to positive, finds the correct maximum value. The ability to correctly formulate the problem is the critical learned skill here.

## C.2 ANALYSIS OF PROCEDURAL VOLATILITY FOR A CRITICAL INSTANCE

To further illustrate how our proposed `CONST` method identifies critical instances, we present a detailed analysis of the **procedural volatility** for a Stratified Sampling problem. This process involves truncating the model's reasoning process at various stages and extracting an answer from each partial thought process. The analysis demonstrates how the model's path to the correct answer is unstable, with its intermediate conclusions fluctuating significantly. This volatility highlights a fragile grasp of the multi-step procedure, which makes this sample a high-value critical instance for fine-tuning.

---

**Procedural Volatility Analysis**

**Full Reasoning Process:**

*To find the number of sophomores, we first need to find the total number of students in the school. We know that there are 450 freshmen and 250 seniors, but we don't know the total number of students (n). However, we can use the information about the stratified random sampling to find the number of sophomores. In stratified random sampling, the sample is divided into subgroups based on the population characteristics (in this case, grade level). The number of students in each subgroup is proportional to the number of students in that subgroup in the population. Let's denote the number of sophomores as x. The total number of students in the sample is 60, and the number of freshmen in the sample is 27. We can set up a proportion to represent the ratio of freshmen in the sample to the total number of freshmen in the population: 27/60 = 450/n We can cross-multiply and solve for n: 27n = 60 \* 450 ⇒ 27n = 27000 ⇒ n = 1000. Now that we know the total number of students (n = 1000), we can find the number of sophomores. The total number of students is the sum of freshmen, sophomores, and seniors: 1000 = 450 + x + 250. Subtract 700 from both sides: 300 = x. So, the number of sophomores is 300.*

---

**Analysis of Truncated Processes:**

- **Stage 1 (25% of Process)**
  - *Truncated Reasoning:* "To find the number of sophomores, we first need to find the total number of students in the school. We know that there are 450 freshmen and 250 seniors... we can use the information about the stratified random sampling... In"
  - *Extracted Answer:* \boxed{173}

- **Stage 2 (50% of Process)**
  - *Truncated Reasoning:* "...the sample is divided into subgroups... The number of students in each subgroup is proportional to the number of students in that subgroup in the population. Let's denote the number of sophomores as x. The total number of students in the sample is 60,"
  - *Extracted Answer:* \boxed{73}

- **Stage 3 (75% of Process)**
  - *Truncated Reasoning:* "...We can set up a proportion... 27/60 = 450/n. We can cross-multiply and solve for n: 27n = 60 \* 450 ⇒ 27n = 27000 ⇒ n = 27000 / 27"
  - *Extracted Answer:* \boxed{1000}

- **Stage 4 (100% of Process)**
  - *Truncated Reasoning:* "...n = 1000. Now that we know the total number of students... we can find the number of sophomores... 1000 = 450 + x + 250. Subtract 700 from both sides: 300 = x. So, the number of sophomores is 300."
  - *Extracted Answer:* \boxed{300}

**Analysis.** The model's reasoning process exhibits high procedural volatility. The extracted answer fluctuates from a hallucination (173), to another incorrect guess (73), to a correct intermediate result mistaken for the final answer (1000), before finally reaching the correct solution (300). This unstable path, despite culminating in a correct answer, reveals a fragile and non-robust understanding of the multi-step problem-solving procedure. This makes the instance a quintessential "critical" or high-value sample for corrective fine-tuning, as it exposes a weakness that simpler, more direct problems would miss.

## C.3   PROMPT TEMPLATE

Our framework utilizes several prompt templates tailored for different tasks, including guiding the model's reasoning process and standardizing its final output for evaluation. The core templates used in our experiments are detailed below.

---

**1. Main Instruction Prompt for Mathematical Reasoning**

This template is appended to every mathematical problem to instruct the model to generate a detailed, step-by-step solution. The final prompt sent to the model is a concatenation of a system prompt, the question, and this instruction.

```
You are a helpful assistant.  You are asked to solve the following
question.  <question>
```

**Let's think step by step and output the final answer within \boxed{}.**

---

**2. Answer Extraction Prompts**

After the model generates a free-form reasoning trace, we use a dedicated extraction prompt to parse the trace and isolate only the final answer. This ensures a standardized format for automated evaluation.

**For General Problems:**

```
You are an expert mathematician and a precise answer extractor.
Your task is to analyze the provided mathematical reasoning and
extract only the final numerical answer. Do not provide any
explanation or preamble. Your final output should ONLY be the
answer enclosed in a \boxed{}.
```

**For MMLU Multiple-Choice Questions:**

```
You are a precise answer extractor. Your task is to analyze the
provided reasoning for a single-choice question and determine
the correct option.

Your final output must be ONLY the letter of the correct option
(e.g., A, B, C, or D) enclosed in a \boxed{}.
```

---

**3. MMLU-Specific Instruction Prompt**

For the MMLU calibration set, which uses a multiple-choice format, a specialized instruction prompt is used to guide the model's response.

```
You will be presented with a single-choice question. Please
analyze the question and the provided options to determine the
single correct answer.

Your final response should be ONLY the letter of the correct
option (e.g., A, B, C, or D) enclosed in a \boxed{}. For
example, if the correct option is B, your response must be
\boxed{B}.
```

# D  ADDITIONAL EXPERIMENTS

## D.1  DIFFERENT SIZES OF CALIBRATION SETS

In this part, we present the model's performance under different sizes of the calibration set (*i.e.*, $m$). We conduct experiments using LLaMA-3.1-8B-Instruct with a fixed annotation budget of $b = 8$. We vary the size $m$ from 256 to 1024, and the results are shown in Table 3. As can be seen from the table, the performance generally improves as the size $m$ increases. Specifically, increasing $m$ from 256 to 1024 leads to a consistent improvement across all datasets. This observation aligns with the intuition that a larger calibration set $\mathcal{D}^{\text{cal}}$ provides a more accurate estimation of the scoring

Table 3: Model's performance under different sizes of the calibration set (*i.e.*, $m$). The best results are marked in **bold**.

| Datasets | AMC23 | MinervaMath | OlympiadBench | MATH500 | AVG |
|---|---|---|---|---|---|
| $m = 256$ | 22.19 | 20.54 | 13.88 | 42.85 | 24.87 |
| $m = 512$ | 23.87 | 22.87 | 16.73 | 45.59 | 27.27 |
| $m = 1024$ (default) | **24.27** | **24.19** | **17.61** | **47.17** | **28.31** |

Table 4: Performance comparison on undergraduate-level STEM problems (OCWCourses) using `LLaMA-3.1-8B-Instruct` with a budget of 8. We report the `avg@32` accuracy. The best results are marked in **bold** and runner-ups with underline.

| Methods | Astronomy | Solid Chem. | Dynamics | AVG |
|---|---|---|---|---|
| NoFinetuning | 7.5 | 15.5 | 26.9 | 16.63 |
| EntSampling | 9.4 | 11.3 | 11.5 | 10.73 |
| CEC | 3.8 | 10.3 | 26.9 | 13.67 |
| `CONST` (ours) | **11.3** | **17.5** | **38.5** | **22.43** |

function's distribution, thereby enhancing the quality of the generated prediction sets $\widehat{C}_{1-\alpha}(X)$. Therefore, we set $m = 1024$ in our main experiments to ensure robust performance.

## D.2 RESULTS ON SCIENTIFIC PROBLEMS

To evaluate the effectiveness of `CONST` in scientific domains, we analyze performance on the OCW-Courses (Lewkowycz et al., 2022) subsets. We report the results of `LLaMA-3.1-8B-Instruct` with a budget of $b = 8$ across three representative science courses: *Introduction to Astronomy*, *Solid State Chemistry*, and *Dynamics and Control*. As shown in Table 4, `CONST` demonstrates substantial improvements over the base model (NoFinetuning) and significantly outperforms other active learning baselines. Notably, classic uncertainty-based methods like EntSampling perform poorly in these hard domains (average accuracy drops to 10.73%), likely because high entropy in these tasks correlates with total model confusion rather than learnable uncertainty. In contrast, `CONST` achieves the highest performance across all three subjects, with an average accuracy of 22.43%, providing a relative improvement of **34.9%** over the original model (16.63%), showing that `CONST` better identifies critical instances. While there may be linguistic noise generated by the model in these domains, `CONST` is less affected and still outperforms baselines.

## D.3 ADDITIONAL BASELINES

To further validate the effectiveness of our proposed framework, we compare `CONST` with two additional baseline methods on `LLaMA-3.1-8B-Instruct` with a budget of 8:

- **SCF** (Self-Consistency Filtering) (Wang et al., 2022): This method evaluates the uncertainty of each instance based on the disagreement among the model's outputs. Specifically, SCF calculates the frequency of the majority answer and selects instances with low self-consistency scores (*i.e.*, high reasoning variance), assuming these instances lie on the decision boundary.

- **EWS** (Entropy-Weighted Sampling) (Beygelzimer et al., 2009): Instead of deterministically selecting the instances with the highest variance, EWS samples instances probabilistically, where the probability of selection is proportional to the predictive entropy.

The results are presented in Table 5. The results show that while SCF and EWS achieve competitive performance (25.87% and 22.14% on average, respectively), surpassing the NoFinetuning baseline, the proposed `CONST` performs better compared to these baselines across all datasets, achieving an average accuracy of 28.31%. These results further demonstrate the superiority of our selection criterion based on conformal prediction considering both procedural and outcome volatility.

Table 5: Performance comparison with additional baselines on `LLaMA-3.1-8B-Instruct` with a budget of 8. We report the `avg@32` (`avg@256` for AMC23) accuracy. The best results are marked in **bold**.

| Methods | AMC23 | MinervaMath | OlympiadBench | MATH500 | AVG |
|---|---|---|---|---|---|
| SCF | 22.83 | 21.63 | 16.36 | 42.64 | 25.87 |
| EWS | 21.38 | 16.14 | 13.71 | 37.31 | 22.14 |
| CONST (ours) | **24.27** | **24.19** | **17.61** | **47.17** | **28.31** |

Table 6: Performance comparison between the likelihood-based baseline (LogProb) and `CONST` using `LLaMA-3.1-8B-Instruct` with a budget of 8. We report the `avg@32` (`avg@256` for AMC23) accuracy. The best results are marked in **bold**.

| Methods | AMC23 | MinervaMath | OlympiadBench | MATH500 | AVG |
|---|---|---|---|---|---|
| LogProb | 22.53 | 23.38 | 15.23 | 46.19 | 26.83 |
| CONST (ours) | **24.27** | **24.19** | **17.61** | **47.17** | **28.31** |

## D.4 ADDITIONAL EXPERIMENTS ON THE SCORING FUNCTION

We also perform ablation studies on the scoring function in conformal prediction. Specifically, we use the model's intrinsic output likelihood as the scoring function:

$$f^{\pi_0}(X, \widehat{Y}) = -\log P(\widehat{Y}, O|X), \tag{14}$$

where $\widehat{Y}$ is the final answer and $O$ is the reasoning trajectory. Note that we also incorporate $O$ since the final answer is often obvious given the reasoning trajectory. The performance comparison is presented in Table 6. We observe that while using log-probability achieves decent performance (26.83% average accuracy), it consistently underperforms `CONST` across all datasets. Specifically, `CONST` surpasses this alternative design by **1.48%** on average, with notable margins on OlympiadBench (+2.38%) and AMC23 (+1.74%). A possible explanation for this result is that the log-probability is prone to linguistic noise (*e.g.*, linguistic unpredictability, rare vocabulary, or stylistic variations), while the proposed method better reflect logical uncertainty. For example, a model might be "surprised" by a token simply because it is an uncommon phrasing, even if the reasoning logic is sound. In contrast, our proposed `CONST` relies on both procedural volatility and outcome volatility, which abstract away from token-level noise to capture higher-level inconsistencies in the reasoning process and the final answer.

## D.5 DIFFERENT CLUSTERING CONFIGURATIONS

We also provide ablation studies of alternative configurations of clustering, where we compare "$b = 8$ clusters, 1 instance in each cluster" (default configuration) and "$b/2 = 4$ clusters, 2 instances in each" (alternative configuration). As can be seen from the results in Table 7, the default configuration is slightly better than the alternative.

## D.6 PASS RATES OF ANSWER SETS

We also conduct a detailed analysis on the pass rates of the generated prediction sets. Specifically, for each test question, we: (*i*) generate $N = 40$ candidate answers using `DeepSeek-R1-Distill-Qwen-1.5B` on the MATH500 dataset, (*ii*) filter them using the scoring function and the threshold $\rho$ derived from the calibration set (with size $m = 100$), and (*iii*) measure the pass rate of the final answer set. We compare `CONST` against two baselines: Self-Consistency and Entropy-based Selection. We set the target error rate $\alpha = 0.2$, implying a target coverage of 80%. The results are presented in Table 8. Note that the average set size of conformal prediction is $k = 3.53$. Since both Self-Consistency and Entropy-Based Selection cannot naturally decide the sizes of the candidate sets for each test question, we fix the size of the candidate sets of these two methods to both $k = 3$ and $k = 4$, for a fair comparison. As can be seen from the results, conformal prediction increases the pass rate of the candidate sets. Conformal prediction can dynamically decide the size of the candidate sets for each question: easy questions will have smaller candidate sets

Table 7: Performance comparison under different clustering configurations of `CONST` using `LLaMA-3.1-8B-Instruct` with a budget of 8. We report the `avg@32` (`avg@256` for AMC23) accuracy. The best results are marked in **bold**.

| Methods | AMC23 | MinervaMath | OlympiadBench | MATH500 | AVG |
|---|---|---|---|---|---|
| alternative | 22.97 | 25.02 | 17.11 | 45.31 | 27.60 |
| default | 24.27 | 24.19 | 17.61 | 47.17 | 28.31 |

to ensure high probabilities of covering the ground truth answer, and hard questions will have larger ones. In this paper, we use the size of the conformal prediction sets to guide sample selection.

Table 8: Pass rate comparison of the candidate sets generated by different selection strategies on MATH500. We use `DeepSeek-R1-Distill-Qwen-1.5B` with $N = 40$ and $\alpha = 0.2$. The best results are marked in **bold**.

| Methods | Pass Rate |
|---|---|
| Self-Consistency ($k = 3$) | 76.75% |
| Entropy-Based Selection ($k = 3$) | 76.50% |
| Self-Consistency ($k = 4$) | 78.50% |
| Entropy-Based Selection ($k = 4$) | 77.75% |
| Conformal Prediction (ours, $k = 3.53$) | **80.75%** |

