# OpenReview forum: "Sample Lottery: Unsupervised Discovery of Critical Instances for LLM Reasoning"
_ICLR.cc/2026/Conference — ICLR 2026 Poster_

### Official Review · Reviewer_fBZw · 2025-10-31

**Soundness:** 3
**Presentation:** 3
**Contribution:** 3
**Rating:** 6
**Confidence:** 3

**Summary:**

This paper focuses on identifying data subsets that achieve performance comparable to using the full dataset. The proposed method, CONST, leverages procedural and outcome volatility to construct conformal prediction sets, enabling unsupervised discovery of critical instances. Experimental results demonstrate that CONST is both efficient and effective.

**Strengths:**

- The problem addressed is important and compelling. Identifying significant samples within datasets can reduce computation, conserve resources, and clarify how data-collection choices influence training.

- Notably, the method requires minimal human annotation, which enhances its generalizability and practical applicability.

**Weaknesses:**

The paper would benefit from clarifying its scope earlier. My understanding is that the method is developed and evaluated specifically for the RLVR setting. To help readers set the right expectations, I suggest explicitly stating this focus in the title and/or abstract. Doing so would also avoid any impression that the approach covers broader RL or non-RL problems.

I also have several additional questions/suggestions, which I listed in the Question part. I did not examine the theoretical part in depth and just provisionally assume its correctness; I’m happy to discuss if other reviewers raise concerns.

**Questions:**

How should  n_p be selected in practice? For each truncated CoT, how many final answers are sampled? Only one or multiple ones?

Does outcome volatility functionally overlap with procedural volatility? In other words, is outcome volatility a special case of procedural volatility?

What are instances? Is that data samples? Like 4 instances means 4 questions in the math dataset.


Could you also report the experimental results where you (i) generate multiple answers per test question, (ii) filter them by the scoring function and the threshold (from Conformal Prediction) to form a candidate subset, and (iii) measure the pass rate of the final answer set? A comparison against self-consistency and entropy-only baselines would be helpful for understanding the motivation for using the calibration set.

In addition, using a calibration set incurs a cost as well: you use m = 1024 instances (vs. 4 or 8 for training). Are these 1024 instances annotated? Finally, the choice of  m likely matters—why fix m=1024? A sensitivity analysis over m would be informative.

Assumptions are central to the validity of your claims; please state them explicitly and tie each result to the assumption it requires: permutation tests and exchangeability. Also, please discuss the practical applicability. Why can we use the MMLU subset as a calibration set for the BM set?

---

> ### Author Response · Authors · 2025-11-21
>
> We are truly grateful for the time you have taken to review our paper, your insightful comments and support. Your positive feedback is incredibly encouraging for us! In the following response, we would like to address your major concern and provide additional clarification.
>
> > Q1. The paper would benefit from clarifying its scope earlier. My understanding is that the method is developed and evaluated specifically for the RLVR setting. To help readers set the right expectations, I suggest explicitly stating this focus in the title and/or abstract. Doing so would also avoid any impression that the approach covers broader RL or non-RL problems.
>
> **A1.** Thank you for the suggestion! We have changed the title to "Sample Lottery: Unsupervised Discovery of Critical Instances in RLVR of LLMs" and the first sentence of the abstract to "Reinforcement Learning with Verifiable Reward (RLVR) has equipped large language models (LLMs) with the capability of reasoning over complicated logical problems through policy optimization."
>
> > Q2. How should n_p be selected in practice? For each truncated CoT, how many final answers are sampled? Only one or multiple ones?
>
> **A2.** Thank you for the question! In our experiments, we set $n_P$ to 20, and we have tried different numbers in Figure 3 (left). In practice, we suggest that $n_P$ should be chosen considering the length of CoT: $n_P$ needs to be large enough to capture the twists and turns in the reasoning trajectories; when it is too large, it may frequently interrupt the logic fragments. For each truncated CoT only one final answer is sampled. We have clarified this in Section 3.2.
>
> > Q3. Does outcome volatility functionally overlap with procedural volatility? In other words, is outcome volatility a special case of procedural volatility?
>
> **A3.** Thank you for the question! Procedural volatility and outcome volatility are different. The former focuses on the volatility within one reasoning path, while the latter focuses on the volatility of multiple reasoning paths.
>
> > Q4. What are instances? Is that data samples? Like 4 instances means 4 questions in the math dataset.
>
> **A4.** Thank you for the question! Yes, the instances refer to the data samples (i.e., the questions in the dataset).
>
> > Q5. Could you also report the experimental results where you (i) generate multiple answers per test question, (ii) filter them by the scoring function and the threshold (from Conformal Prediction) to form a candidate subset, and (iii) measure the pass rate of the final answer set? A comparison against self-consistency and entropy-only baselines would be helpful for understanding the motivation for using the calibration set.
>
> **A5.** Thank you for the suggestion! We have provided results on MATH500 by: (i) generating $N=40$ candidate answers using DeepSeek-R1-Distill-Qwen-1.5B on the MATH500 dataset, (ii) filtering them using the scoring function and the threshold $\rho$ derived from the calibration set (with size $m=100$), and (iii) measuring the pass rate of the final answer set. We compare CONST against two baselines: Self-Consistency and Entropy-Based Selection. We set the target error rate $\alpha=0.2$, implying a target coverage of 80\%. The results are shown below. (We have also added these results in Appendix D.6 of the revised paper.) Note that the average set size of conformal prediction is $k=3.53$. Since both Self-Consistency and Entropy-Based Selection cannot naturally decide the sizes of the candidate sets for each test question, we fix the size of the candidate sets of these two methods to both $k=3$ and $k=4$, for a fair comparison. As can be seen from the results, conformal prediction increases the pass rate of the candidate sets. Conformal prediction can dynamically decide the size of the candidate sets for each question: easy questions will have smaller candidate sets to ensure high probabilities of covering the ground truth answer, and hard questions will have larger ones. In this paper, we use the size of the conformal prediction sets to guide sample selection.
>
> | Methods                          | Pass Rate |
> |----------------------------------|---------|
> | Self-Consistency ($k = 3$)         | 76.75%  |
> | Entropy-Based Selection ($k = 3$$)  | 76.50%  |
> | Self-Consistency ($k = 4$)         | 78.50%  |
> | Entropy-Based Selection ($k = 4$)  | 77.75%  |
> | **CONST (ours, $k = 3.53$)**       | **80.75%** |

---

> ### Author Response · Authors · 2025-11-21
>
> > Q6. In addition, using a calibration set incurs a cost as well: you use m = 1024 instances (vs. 4 or 8 for training). Are these 1024 instances annotated? Finally, the choice of m likely matters—why fix m=1024? A sensitivity analysis over m would be informative.
>
> **A6.** Thank you for the question! Yes, the calibration set is annotated. In practice, we can use existing datasets that are already annotated (e.g., MMLU) to avoid the cost of annotating the calibration set.
>
> We have provided results of the model's performance under different sizes of the calibration set (i.e., $m$). We conduct experiments using LLaMA-3.1-8B-Instruct with a fixed annotation budget of $b=8$, and vary the size $m$ from 256 to 1024, and the results are shown below. (We have also added these results in Appendix D.1 of the revised paper.)
>
> | Datasets              | AMC23  | MinervaMath | OlympiadBench | MATH500 | AVG    |
> |-----------------------|--------|-------------|---------------|---------|--------|
> | $m = 256$             | 22.19  | 20.54       | 13.88         | 42.85   | 24.87  |
> | $m = 512$             | 23.87  | 22.87       | 16.73         | 45.59   | 27.27  |
> | **$m = 1024$ (default)** | **24.27** | **24.19**   | **17.61**     | **47.17** | **28.31** |
>
>
> As can be seen from the table, the performance generally improves as the size $m$ increases, since a larger calibration set $\mathcal{D}^{\text{cal}}$ provides a more accurate estimation of the scoring function's distribution, thereby enhancing the quality of the generated prediction sets $\widehat{C}_{1-\alpha}(X)$. Note that the size of the calibration set is more of a statistical requirement (to ensure accurate score distribution), and that we have shown the method's robustness to the choice of the calibration set in Section 4.4, which means that we can use datasets that are already annotated (e.g., MMLU) as the calibration set in practice.
>
> > Q7. Assumptions are central to the validity of your claims; please state them explicitly and tie each result to the assumption it requires: permutation tests and exchangeability. Also, please discuss the practical applicability. Why can we use the MMLU subset as a calibration set for the BM set?
>
> **A7.** Thank you for the suggestion!
>
> The proof of our Theorem 3.1 consists of two main components: (1) Convergence of Gradient Descent (Appendix A.1) and (2) Generalization of GRPO (Appendix A.2). The first component, regarding convergence, is a deterministic proof under Assumptions 3.2 and 3.3. It does not involve any assumptions about sequential dependencies or correlations. For the second component on generalization, we require a group-level independence assumption—specifically, independence between groups (e.g., different questions), rather than independence within sequences (e.g., tokens).
>
> As for the second part, a possible reason why we can use MMLU as a calibration set for BM is that a specific LLM and the scoring function may share similar properties or behaviors across different datasets, and thus the threshold obtained from one dataset can be transferred to another. Additionally, our method mainly concerns the sizes of the conformal prediction sets. The order of these sizes may be preserved when using a slightly different threshold.
>
>
> We will properly incorporate your suggestions into our revised version. Thanks again for appreciating our work and for your constructive suggestions. Please let us know if you have further questions.

---

### Official Review · Reviewer_d26V · 2025-11-01

**Soundness:** 3
**Presentation:** 2
**Contribution:** 3
**Rating:** 6
**Confidence:** 3

**Summary:**

The paper investigates whether a very small subset of training problems can drive reinforcement-learning-with-verifiable-reward (RLVR) fine-tuning of large language models to the same accuracy obtained from the fully-annotated corpus. It fuses procedural volatility (how unstable an answer is to reasoning truncation) and outcome volatility (how inconsistent full answers are across rollouts) into a single sample-utility score using conformal prediction. This mechanism is intuitive, computationally light compared to full uncertainty modeling, and agnostic to the base model or reward function—making it broadly applicable.

**Strengths:**

1. If validated at larger scale, CONST could substantially reduce reward annotation costs in reasoning RL pipelines, which is an increasingly important direction for sustainable LLM alignment.

2. Novel formulation: Using conformal prediction to combine procedural and outcome uncertainty for sample scoring is an elegant, model-agnostic idea.

3. The figures and algorithms are cleanly presented; the method’s intuition is easy to follow even for non-specialists.

**Weaknesses:**

1. Similar ideas have appeared under names like self-consistency filtering, or uncertainty-guided selection. CONST’s originality mainly comes from framing these within conformal prediction and the RLVR objective.

2. The “ε-approximate lottery-sample” assumption is interesting but unverifiable. The authors don’t measure ε or show gradient proximity, so the bound doesn’t really illuminate why CONST works.

3. Procedural and outcome volatility both depend on stochastic decoding variance. Without separating linguistic noise from reasoning uncertainty, the method could mis-rank samples for harder domains.

4. Compared mainly to classification-style active learning. RL- or reasoning-specific selection strategies (e.g., self-consistency filtering, entropy-weighted sampling, or value-driven selection) are absent.

**Questions:**

1. Why not incorporate log P(Y | X) from π₀ into fπ₀(X,Y)? Did likelihood-based scoring perform worse?

2. Which embedding and K were used? How sensitive is CONST to these settings? Provide ablations with identical budgets but varying clustering to isolate diversity effects.

3. Could CONST operate in an active loop where the fine-tuned policy re-selects new samples? Any preliminary results?

---

> ### Author Response · Authors · 2025-11-21
>
> We are truly grateful for the time you have taken to review our paper, your insightful comments and support. Your positive feedback is incredibly encouraging for us! In the following response, we would like to address your major concern and provide additional clarification.
>
> > Q1. Similar ideas have appeared under names like self-consistency filtering, or uncertainty-guided selection. CONST’s originality mainly comes from framing these within conformal prediction and the RLVR objective.
>
> **A1.** Thank you for the comment! The novelty of this work is three-fold:
> * **Different motivation**: This work measures the value of training samples under the framework of conformal prediction, offering a probabilistic approach of finding important instances.
> * **Different technique**: We consider both procedural volatility and outcome volatility of LLMs' reasoning trajectories.
> * **Different scenario**: This work focuses on reinforcement learning with verifiable reward (RLVR) on LLMs, while previous data valuation efforts mainly focus on image classification with smaller models.
> * **Theoretical support**: We provide theoretical analysis of our framework, showing the method's ability to approximate the optimal parameter setup with the full dataset.
>
> > Q2. The “ε-approximate lottery-sample” assumption is interesting but unverifiable. The authors don’t measure ε or show gradient proximity, so the bound doesn’t really illuminate why CONST works.
>
> **A2.** Thank you for the comment! We now provide a high-level justification for why Assumption 3.1 posits that the question set $\mathcal Q’$ selected by the **CONST** method is an $\epsilon$-approximation of the full training set $\mathcal Q\triangleq\{X_1, X_2, \dots, X_N\}$.
>
> First, note that if we ignore the regularization term $\lambda\cdot\mathcal D_\text{KL}(\pi_\theta || \pi_0)$( i.e., let $\lambda\rightarrow 0$), we can show that, without loss of generality, the following equality holds for any subset $S \subset \mathcal Q$,
>
>  $\nabla\hat{\mathcal{L}}^{S}=-\frac{1}{n|S|} \sum_{X\in S}\sum_{i=1}^{n}\frac{\nabla\pi_\theta(O_i | X)}{\pi_{\theta'}(O_i | X)} a_i$
>
> where$a_{i}\triangleq\frac{r_i-mean(\{r_j\})}{std(\{r_j\})}$ is the advantage calculated based on relative rewards.
>
> Then, we decompose the gradient $\nabla\hat{\mathcal{L}}^{\mathcal Q}$ of  full training set $ \mathcal Q$.
>
>  Before going into the details, we introduce the notations $\mathcal Q_{\delta}\triangleq(X\in\mathcal{Q} : |a_{i}|\le\delta,\forall i\in[n])$ and $\mathcal{Q}_{>\delta}$=
>
> $\mathcal{Q}\setminus\mathcal{Q}_{\delta}$. Subsequently, we have that
>
> $\nabla\hat{\mathcal{L}}^{\mathcal Q}=- \frac{1}{n|\mathcal{Q}|} \sum_{X\in \mathcal Q}\sum_{i=1}^{n}
>         \frac{\nabla\pi_\theta(O_i | X)}{\pi_{\theta'}(O_i | X)} a_i=-\frac{1}{n|\mathcal Q|} \sum_{X\in Q_{\delta}}\sum_{i=1}^{n}
>         \frac{\nabla\pi_\theta(O_i | X)}{\pi_{\theta'}(O_i | X)} a_i-\frac{1}{n|\mathcal Q|} \sum_{X\in\mathcal Q_{>\delta}}\sum_{i=1}^{n}
>         \frac{\nabla\pi_\theta(O_i | X)}{\pi_{\theta'}(O_i | X)} a_i$
>
> It is worth noting that, when all $a_{i}<\delta$ and the ratio $\frac{\nabla\pi_\theta(O_i | X)}{\pi_{\theta'}(O_i | X)}$ is bounded, we obtain:
>
> $||\frac{1}{n|\mathcal Q|} \sum_{X\in Q_{\delta}}\sum_{i=1}^{n}
>         \frac{\nabla\pi_\theta(O_i | X)}{\pi_{\theta'}(O_i | X)} a_i||\le \mathcal{O}(\frac{|\mathcal{Q}_{\delta}|}{|\mathcal Q|}\delta)$ which implies
>
> $\nabla\hat{\mathcal{L}}^{\mathcal Q}\approx -\mathcal{O}(\frac{| Q_{\delta}|}{|\mathcal Q|}\delta)-\frac{1}{n|\mathcal Q|} \sum_{X\in Q_{>\delta}}\sum_{i=1}^{n}  \frac{\nabla\pi_\theta(O_i | X)}{\pi_{\theta'}(O_i | X)} a_i$.
>
> In other words, if $\delta=\mathcal{O}(\epsilon)$, we can guarantee that
>
> $||\nabla\hat{\mathcal{L}}^{\mathcal Q}-(-\frac{1}{n|\mathcal Q|} \sum_{X\in Q_{>\delta}}\sum_{i=1}^{n}  \frac{\nabla\pi_\theta(O_i | X)}{\pi_{\theta'}(O_i | X)} a_i)||\le\mathcal{O}(\epsilon) $.
>
> Finally, we analyze the remaining term. The key observation is that our final scoring function $f^{\pi_0}(X,\widehat Y)$ incorporates the negative frequency component $f_\text{NF}(X,\widehat Y)$. Therefore, by appropriately controlling $\delta$, we can ensure that $\mathcal{Q}’\subseteq\mathcal Q_{>\delta}$, namely, for any data $X\in\mathcal{Q}’$, there exist an $a_{i}>\delta$.  Furthermore, by carefully adjusting the quantile threshold, we can achieve
> $||\frac{1}{n|\mathcal Q|} \sum_{X\in\mathcal Q’}\sum_{i=1}^{n}  \frac{\nabla\pi_\theta(O_i | X)}{\pi_{\theta'}(O_i | X)} a_i-\frac{1}{n|\mathcal Q|} \sum_{X\in\mathcal Q_{>\delta}}\sum_{i=1}^{n}  \frac{\nabla\pi_\theta(O_i | X)}{\pi_{\theta'}(O_i | X)} a_i||\le\mathcal{O}(\epsilon)$ which ultimately yields
>
> $||\nabla\hat{\mathcal{L}}^{\mathcal{Q}'}(\theta)-\nabla\hat{\mathcal{L}}^{\mathcal{Q}}(\theta)||_{2}\le\mathcal{O}(\epsilon)$,
>
> where $\nabla\hat{\mathcal{L}}^{\mathcal{Q}'}(\theta)=-\frac{1}{n|\mathcal Q|} \sum_{X\in\mathcal Q'}\sum_{i=1}^{n}  \frac{\nabla\pi_\theta(O_i | X)}{\pi_{\theta'}(O_i | X)} a_i$.

---

> ### Author Response · Authors · 2025-11-21
>
> > Q3. Procedural and outcome volatility both depend on stochastic decoding variance. Without separating linguistic noise from reasoning uncertainty, the method could mis-rank samples for harder domains.
>
> **A3.** Thank you for the comment! We have provided results on harder scientific domains below (the results have also been added to Appendix D.2). Specifically, we evaluate the model's performance on three scientific domains: "Introduction to Astronomy", "Solid State Chemistry", and "Dynamics and Control". As can be seen from the results, CONST outperforms baselines by a large margin. While there may be linguistic noise generated by the model in these domains, CONST is less affected and still outperforms baselines.
>
> | Methods          | Astronomy | Solid Chem. | Dynamics | AVG    |
> |------------------|-----------|-------------|----------|--------|
> | NoFinetuning     | 7.5       | 15.5        | 26.9     | 16.63  |
> | EntSampling      | 9.4       | 11.3        | 11.5     | 10.73  |
> | CEC              | 3.8       | 10.3        | 26.9     | 13.67  |
> | **CONST (ours)** | **11.3**  | **17.5**    | **38.5** | **22.43** |
>
>
> > Q4. Compared mainly to classification-style active learning. RL- or reasoning-specific selection strategies (e.g., self-consistency filtering, entropy-weighted sampling, or value-driven selection) are absent.
>
> **A4.** Thank you for the suggestion! Here, we have provided additional results of self-consistency filtering (SCF) and entropy-weighted sampling (EWS). The results have also been added to Appendix E.4. We compare CONST with the two baselines on LLaMA-3.1-8B-Instruct with a budget of 8, and the results are shown below. (We have also added them in Appendix D.3). As can be seen from the results, the proposed CONST outperforms these baselines, showing the effectiveness of the method.
>
> | Methods          | AMC23  | MinervaMath | OlympiadBench | MATH500 | AVG    |
> |------------------|--------|-------------|---------------|---------|--------|
> | SCF              | 22.83  | 21.63       | 16.36         | 42.64   | 25.87  |
> | EWS              | 21.38  | 16.14       | 13.71         | 37.31   | 22.14  |
> | **CONST (ours)** | **24.27** | **24.19**   | **17.61**     | **47.17** | **28.31** |
>
>
> > Q5. Why not incorporate log P(Y | X) from π₀ into fπ₀(X,Y)? Did likelihood-based scoring perform worse?
>
> **A5.** Thank you for the question! We have provided an experiment that uses $\log P(Y|X)$ as the scoring function. Specifically, we set the scoring function as $f^{\pi_0}(X,\widehat Y) = -\log P(\widehat Y, O|X)$, where $\widehat Y$ is the final answer and $O$ is the reasoning trajectory. Note that we also incorporate $O$ since the final answer is often obvious given the reasoning trajectory. The performance comparison is shown below, and we have also added the experiment in Appendix D.4.
>
> | Methods          | AMC23  | MinervaMath | OlympiadBench | MATH500 | AVG    |
> |------------------|--------|-------------|---------------|---------|--------|
> | LogProb          | 22.53  | 23.38       | 15.23         | 46.19   | 26.83  |
> | **CONST (ours)** | **24.27** | **24.19**   | **17.61**     | **47.17** | **28.31** |
>
> The results show that the scoring function used in CONST is better than the alternative one using log-probability. A possible explanation for this result is that the log-probability is prone to linguistic noise, while the proposed method better reflects logical uncertainty.
>
>
> > Q6. Which embedding and K were used? How sensitive is CONST to these settings? Provide ablations with identical budgets but varying clustering to isolate diversity effects.
>
> **A6.** Thank you for the question! We use Sentence-BERT [1] to obtain the embeddings of the input queries, and use K-means algorithm to obtain the clusters. The number of clusters is set to $b$, which is the budget of annotation. We have added the details about clustering in Section 4.1. We have also provided ablation studies of alternative configurations of clustering, where we compare "8 clusters, 1 in each" (default) and "4 clusters, 2 in each" (alternative). We have provided the results in Appendix D.5 and we also list the results below:
>
> Exp|AMC23|MinervaMath|OlympiadBench|MATH500|AVG
> -|-|-|-|-|-
> default|24.27|24.19|17.61|47.17|28.31
> alternative|22.97|25.02|17.11|45.31|27.60
>
> [1] Reimers, Nils, and Iryna Gurevych. "Sentence-BERT: Sentence Embeddings using Siamese BERT-Networks." In EMNLP 2019.

---

> ### Author Response · Authors · 2025-11-21
>
> > Q7. Could CONST operate in an active loop where the fine-tuned policy re-selects new samples? Any preliminary results?
>
> **A7.** Thank you for the question! CONST can operate in an active loop, and we have provided the results below and in Section 4.4. From the results, we can see that as the policy evolves, CONST can effectively identify critical instances, achieving improvement in an active loop.
>
> | Rounds   | AMC23  | MinervaMath | OlympiadBench | MATH500 | AVG    |
> |-------------|--------|-------------|---------------|---------|--------|
> | Round 1 | 24.27  | 24.19       | 17.61         | 47.17   | 28.31  |
> | Round 2 | 25.29  | 27.31       | 18.73         | 51.24   | 30.64  |
>
>
> We will properly incorporate your suggestions into our revised version. Thanks again for appreciating our work and for your constructive suggestions. Please let us know if you have further questions.

---

### Official Review · Reviewer_H6dA · 2025-11-01

**Soundness:** 4
**Presentation:** 3
**Contribution:** 4
**Rating:** 8
**Confidence:** 3

**Summary:**

This paper proposes CONST, a framework for identifying lottery-winning samples that are most critical for RLVR in LLMs. Instead of requiring full annotations for all training data, CONST selects a very small subset of informative instances without access to ground-truth answers. It combines two complementary measures: procedural volatility (instability of reasoning paths) and outcome volatility (variability in final answers), and leverages conformal prediction to quantify uncertainty through the size of prediction sets. Samples with higher uncertainty are selected for annotation and used for RLVR optimization. Theoretical analysis under the lottery sample hypothesis shows CONST can approximate the optimal policy, and experiments on several mathematical reasoning benchmarks demonstrate that CONST achieves near full-dataset performance with less than 0.5% of annotated samples, outperforming existing active learning baselines.

**Strengths:**

1. The method is valuable because it helps us understand which samples truly improve model performance and which do not, providing insight into data efficiency for RL-based reasoning.

2. The experiments are comprehensive and convincing, covering multiple models, datasets, and ablation settings to clearly show the method’s effectiveness.

3. The paper is easy to follow and well organized, with a clear narrative from motivation to theory and experiments, making complex ideas accessible.

**Weaknesses:**

1. I want to know whether CONST is suitable for logic reasoning datasets with discrete or small answer spaces such as multiple-choice tasks with only four options, where outcome volatility may be artificially low and conformal prediction less informative.

2. Theoretical results justify that an optimal subset can approximate full-data training, but the analysis stops short of proving that CONST reliably finds such a subset, the connection between the proposed selection criterion and the theoretical gradient proximity assumption remains heuristic.

3. Is the method sensitive to the number of instances in the calibration dataset?

4. The ablation (V2) skips the clustering step in Algorithm 1, but the paper never explains how clustering is done—what features or metrics are used and how the number of clusters is chosen.

**Questions:**

See weakness.

---

> ### Author Response · Authors · 2025-11-21
>
> We are truly grateful for the time you have taken to review our paper, your insightful comments and support. Your positive feedback is incredibly encouraging for us! In the following response, we would like to address your major concern and provide additional clarification.
>
> > Q1. I want to know whether CONST is suitable for logic reasoning datasets with discrete or small answer spaces such as multiple-choice tasks with only four options, where outcome volatility may be artificially low and conformal prediction less informative.
>
> **A1.** Thank you for the question! When dealing with questions with very small answer spaces, the volatility can be low, and the conformal prediction could be less informative. To mitigate this effect, a potential solution is to remove the choices when computing procedural/outcome volatility.
>
> > Q2. Theoretical results justify that an optimal subset can approximate full-data training, but the analysis stops short of proving that CONST reliably finds such a subset, the connection between the proposed selection criterion and the theoretical gradient proximity assumption remains heuristic.
>
> **A2.** Thank you for this insightful comment. We now provide a high-level justification for why Assumption 3.1 posits that the question set $\mathcal Q’$ selected by the **CONST** method is an $\epsilon$-approximation of the full training set $\mathcal Q\triangleq\{X_1, X_2, \dots, X_N\}$.
>
> First, note that if we ignore the regularization term $\lambda\cdot\mathcal D_\text{KL}(\pi_\theta || \pi_0)$( i.e., let $\lambda\rightarrow 0$), we can show that, without loss of generality, the following equality holds for any subset $S \subset \mathcal Q$,
>
>  $\nabla\hat{\mathcal{L}}^{S}=-\frac{1}{n|S|} \sum_{X\in S}\sum_{i=1}^{n}\frac{\nabla\pi_\theta(O_i | X)}{\pi_{\theta'}(O_i | X)} a_i$
>
> where$a_{i}\triangleq\frac{r_i-mean(\{r_j\})}{std(\{r_j\})}$ is the advantage calculated based on relative rewards.
>
> Then, we decompose the gradient $\nabla\hat{\mathcal{L}}^{\mathcal Q}$ of  full training set $ \mathcal Q$.
>
>  Before going into the details, we introduce the notations $\mathcal Q_{\delta}\triangleq(X\in\mathcal{Q} : |a_{i}|\le\delta,\forall i\in[n])$ and $\mathcal{Q}_{>\delta}$=
>
> $\mathcal{Q}\setminus\mathcal{Q}_{\delta}$. Subsequently, we have that
>
> $\nabla\hat{\mathcal{L}}^{\mathcal Q}=- \frac{1}{n|\mathcal{Q}|} \sum_{X\in \mathcal Q}\sum_{i=1}^{n}
>         \frac{\nabla\pi_\theta(O_i | X)}{\pi_{\theta'}(O_i | X)} a_i=-\frac{1}{n|\mathcal Q|} \sum_{X\in Q_{\delta}}\sum_{i=1}^{n}
>         \frac{\nabla\pi_\theta(O_i | X)}{\pi_{\theta'}(O_i | X)} a_i-\frac{1}{n|\mathcal Q|} \sum_{X\in\mathcal Q_{>\delta}}\sum_{i=1}^{n}
>         \frac{\nabla\pi_\theta(O_i | X)}{\pi_{\theta'}(O_i | X)} a_i$
>
> It is worth noting that, when all $a_{i}<\delta$ and the ratio $\frac{\nabla\pi_\theta(O_i | X)}{\pi_{\theta'}(O_i | X)}$ is bounded, we obtain:
>
> $||\frac{1}{n|\mathcal Q|} \sum_{X\in Q_{\delta}}\sum_{i=1}^{n}
>         \frac{\nabla\pi_\theta(O_i | X)}{\pi_{\theta'}(O_i | X)} a_i||\le \mathcal{O}(\frac{|\mathcal{Q}_{\delta}|}{|\mathcal Q|}\delta)$ which implies
>
> $\nabla\hat{\mathcal{L}}^{\mathcal Q}\approx -\mathcal{O}(\frac{| Q_{\delta}|}{|\mathcal Q|}\delta)-\frac{1}{n|\mathcal Q|} \sum_{X\in Q_{>\delta}}\sum_{i=1}^{n}  \frac{\nabla\pi_\theta(O_i | X)}{\pi_{\theta'}(O_i | X)} a_i$.
>
> In other words, if $\delta=\mathcal{O}(\epsilon)$, we can guarantee that
>
> $||\nabla\hat{\mathcal{L}}^{\mathcal Q}-(-\frac{1}{n|\mathcal Q|} \sum_{X\in Q_{>\delta}}\sum_{i=1}^{n}  \frac{\nabla\pi_\theta(O_i | X)}{\pi_{\theta'}(O_i | X)} a_i)||\le\mathcal{O}(\epsilon) $.
>
> Finally, we analyze the remaining term. The key observation is that our final scoring function $f^{\pi_0}(X,\widehat Y)$ incorporates the negative frequency component $f_\text{NF}(X,\widehat Y)$. Therefore, by appropriately controlling $\delta$, we can ensure that $\mathcal{Q}’\subseteq\mathcal Q_{>\delta}$, namely, for any data $X\in\mathcal{Q}’$, there exist an $a_{i}>\delta$.  Furthermore, by carefully adjusting the quantile threshold, we can achieve
> $||\frac{1}{n|\mathcal Q|} \sum_{X\in\mathcal Q’}\sum_{i=1}^{n}  \frac{\nabla\pi_\theta(O_i | X)}{\pi_{\theta'}(O_i | X)} a_i-\frac{1}{n|\mathcal Q|} \sum_{X\in\mathcal Q_{>\delta}}\sum_{i=1}^{n}  \frac{\nabla\pi_\theta(O_i | X)}{\pi_{\theta'}(O_i | X)} a_i||\le\mathcal{O}(\epsilon)$ which ultimately yields
>
> $||\nabla\hat{\mathcal{L}}^{\mathcal{Q}'}(\theta)-\nabla\hat{\mathcal{L}}^{\mathcal{Q}}(\theta)||_{2}\le\mathcal{O}(\epsilon)$,
>
> where $\nabla\hat{\mathcal{L}}^{\mathcal{Q}'}(\theta)=-\frac{1}{n|\mathcal Q|} \sum_{X\in\mathcal Q'}\sum_{i=1}^{n}  \frac{\nabla\pi_\theta(O_i | X)}{\pi_{\theta'}(O_i | X)} a_i$.
>
> This rigorous decomposition demonstrates how **CONST**  selection criterion connects to the gradient proximity assumption, providing the theoretical foundation for our approximation guarantee.

---

> ### Author Response · Authors · 2025-11-21
>
> > Q3. Is the method sensitive to the number of instances in the calibration dataset?
>
> **A3.** Thank you for the question! We have provided results of the model's performance under different sizes of the calibration set (i.e., $m$). We conduct experiments using LLaMA-3.1-8B-Instruct with a fixed annotation budget of $b=8$. We vary the size $m$ from 256 to 1024, and the results are shown below. (We have also added these results in Appendix D.1 of the revised paper.)
>
> | Datasets              | AMC23  | MinervaMath | OlympiadBench | MATH500 | AVG    |
> |-----------------------|--------|-------------|---------------|---------|--------|
> | $m = 256$             | 22.19  | 20.54       | 13.88         | 42.85   | 24.87  |
> | $m = 512$             | 23.87  | 22.87       | 16.73         | 45.59   | 27.27  |
> | **$m = 1024$ (default)** | **24.27** | **24.19**   | **17.61**     | **47.17** | **28.31** |
>
>
> As can be seen from the table, the performance generally improves as the size $m$ increases, since a larger calibration set $\mathcal{D}^{\text{cal}}$ provides a more accurate estimation of the scoring function's distribution, thereby enhancing the quality of the generated prediction sets $\widehat{C}_{1-\alpha}(X)$. Note that the size of the calibration set is more of a statistical requirement (to ensure accurate score distribution), and that we have shown the method's robustness to the choice of the calibration set in Section 4.4, which means that we can use datasets that are already annotated (e.g., MMLU) as the calibration set in practice.
>
>
> > Q4. The ablation (V2) skips the clustering step in Algorithm 1, but the paper never explains how clustering is done—what features or metrics are used and how the number of clusters is chosen.
>
> **A4.** Thank you for the comment! We use Sentence-BERT [1] to obtain the embeddings of the input queries, and use the K-means algorithm to obtain the clusters. The number of clusters is set to $b$, which is the budget of annotation. We have added more details in the revised version (Section 4.1 Implementation Details).
>
> [1] Reimers, Nils, and Iryna Gurevych. "Sentence-BERT: Sentence Embeddings using Siamese BERT-Networks." In EMNLP 2019.
>
>
> We will properly incorporate your suggestions into our revised version. Thanks again for appreciating our work and for your constructive suggestions. Please let us know if you have further questions.

---

> > ### Comment · Reviewer_H6dA · 2025-11-28
> > **Response to authors**
> >
> > Thanks for the authors’ response. The reply addresses most of my concerns, and I’ve decided to keep my original scores.

---

### Official Review · Reviewer_rXws · 2025-11-07

**Soundness:** 3
**Presentation:** 2
**Contribution:** 1
**Rating:** 4
**Confidence:** 4

**Summary:**

This paper describes a new data subset selection method called CONST. CONST identifies the critical instances from training data by measuring their procedural volatility (variations in the reasoning chain) and outcome volatility (inconsistencies in the final answer), combining these metrics using conformal prediction to determine sample importance. Experimental results demonstrate that training on the small subset selected by CONST achieves comparable performance to full-dataset training, showcasing its effectiveness for data-efficient policy optimization.

**Strengths:**

- The problem is important and interesting. It is good that the proposed method requires minimal annotation from experts and also does not use any LLM annotations.

- The authors analyse their algorithm theoretically.

- The authors show that using a small number of examples gives almost as good a performance as the whole dataset.

**Weaknesses:**

- The main drawback is that the authors do not discuss the whole of data subset selection and valuation literature. Example papers include:

Paul, Mansheej, Surya Ganguli, and Gintare Karolina Dziugaite. "Deep learning on a data diet: Finding important examples early in training." Advances in neural information processing systems 34 (2021): 20596-20607.

Guo, Chengcheng, Bo Zhao, and Yanbing Bai. "Deepcore: A comprehensive library for coreset selection in deep learning." In International Conference on Database and Expert Systems Applications, pp. 181-195. Cham: Springer International Publishing, 2022.

Das, Soumi, Manasvi Sagarkar, Suparna Bhattacharya, and Sourangshu Bhattacharya. "CheckSelect: Online Checkpoint Selection for Flexible, Accurate, Robust, and Efficient Data Valuation." IEEE Transactions on Artificial Intelligence (2024).

- I am not sure about the validity of assumption 3. The authors cite another paper, calling it a standard assumption. However, in my opinion, this assumption is very strong and is often invalid. Also, I am not sure if the proof is novel under this assumption.

- The authors did not show results on bigger reasoning models.

**Questions:**

Why are the authors not reporting results on qwen 2.5 7 b or qwen 3 8b ?

---

> ### Author Response · Authors · 2025-11-21
>
> We are truly grateful for the time you have taken to review our paper and your insightful review. Here we address your comments in the following.
>
> > Q1. The main drawback is that the authors do not discuss the whole of data subset selection and valuation literature. Example papers include:
> >
> > Paul, Mansheej, Surya Ganguli, and Gintare Karolina Dziugaite. "Deep learning on a data diet: Finding important examples early in training." Advances in neural information processing systems 34 (2021): 20596-20607.
> >
> > Guo, Chengcheng, Bo Zhao, and Yanbing Bai. "Deepcore: A comprehensive library for coreset selection in deep learning." In International Conference on Database and Expert Systems Applications, pp. 181-195. Cham: Springer International Publishing, 2022.
> >
> > Das, Soumi, Manasvi Sagarkar, Suparna Bhattacharya, and Sourangshu Bhattacharya. "CheckSelect: Online Checkpoint Selection for Flexible, Accurate, Robust, and Efficient Data Valuation." IEEE Transactions on Artificial Intelligence (2024).
>
> **A1.** Thank you for sharing these excellent works with us! We have added discussion about data subset selection and valuation literature including these works in the revised version (in Section 5). Specifically, Paul et al. [1] propose the Gradient Normed (GraNd) and the Error L2-Norm (EL2N) scores to select important examples very early in training, reducing the size of the training set effectively. Guo et al. [2] provide a comprehensive code library for selecting a subset of the most informative training samples, and conduct extensive evaluations of popular methods. Das et al. [3] propose CheckSelect, a flexible, accurate, robust, and efficient method for extracting the high-value subsets. Extensive experiments show that CheckSelect outperforms state-of-the-art baselines while remaining efficient.
>
> Nevertheless, many works on data subset selection and valuation assume complete annotation of the training set by computing the loss function or the gradient. By comparison, this work aims to find important training data (the critical instances) **without annotation**, and then only annotate important samples.
>
> [1] Paul, Mansheej, Surya Ganguli, and Gintare Karolina Dziugaite. "Deep learning on a data diet: Finding important examples early in training." Advances in neural information processing systems 34 (2021): 20596-20607.
>
> [2] Guo, Chengcheng, Bo Zhao, and Yanbing Bai. "Deepcore: A comprehensive library for coreset selection in deep learning." In International Conference on Database and Expert Systems Applications, pp. 181-195. Cham: Springer International Publishing, 2022.
>
> [3] Das, Soumi, Manasvi Sagarkar, Suparna Bhattacharya, and Sourangshu Bhattacharya. "CheckSelect: Online Checkpoint Selection for Flexible, Accurate, Robust, and Efficient Data Valuation." IEEE Transactions on Artificial Intelligence (2024).
>
> > Q2. I am not sure about the validity of assumption 3. The authors cite another paper, calling it a standard assumption. However, in my opinion, this assumption is very strong and is often invalid. Also, I am not sure if the proof is novel under this assumption.
>
> **A2.** Thank you for the comment! Due to space constraints,  we only briefly cite prior work to state that Assumptions 3.2 and 3.3 are standard assumptions in the original version. We now elaborate on this in detail.
>
> - **Validity of Assumption 3.3**:  It is worth noting that the Polyak-Lojasiewicz(PL) condition has been extensively validated in deep learning literature. For instance, (Yuan et al., 2019) have also furnished empirical evidence of the PL condition's presence during the training of deep neural networks. Moreover, (Allen-Zhu et al., 2019) offer theoretical proof of its ability to guarantee linear convergence of gradient-based methods in non-convex optimization.
> -  **Validity of Assumption 3.2**: First, let us recall a conclusion of the Vitali covering theorem(Evans, 2018), that is, "Lipschitz continuity implies differentiability almost everywhere". This means we only require the objective function $\hat{\mathcal{L}}_\text{GRPO}^{Q}(\theta)$ to have bounded second-order differentiability to prove $L$-smoothness. Such boundedness is very natural for many deep architectures.
> - **Novelty of our Proofs**:  It is worth noting that we additionally introduce the notions of ergodic Markov properties and mixing time to characterize the decision-making process of large models, which makes, compared with previous results, the proof of Theorem 3.1 not a simple repetition. We need to establish a new generalization error analysis under GRPO. To the best of our knowledge, no prior work has done this before.
>
> **References**
>
>  Allen-Zhu, Z., Li, Y., and Song, Z. A convergence theory for
> deep learning via over-parameterization. In ICML,2019.
>
> Yuan, Z., Yan, Y., Jin, R., and Yang, T. Stagewise training
> accelerates convergence of testing error over sgd. NeurIPS 2019.
>
> Lawrence Evans. Measure theory and fine properties of functions. Routledge, 2018.

---

> ### Author Response · Authors · 2025-11-21
>
> > Q3. The authors did not show results on bigger reasoning models. Why are the authors not reporting results on qwen 2.5 7 b or qwen 3 8b ?
>
> **A3.** Thank you for the suggestion! We have provided results on Qwen 2.5 7B below (and in Table 1 of the revised paper). As can be seen from the results, the proposed CONST consistently outperforms baselines across different datasets. With a budget of $b=8$, CONST achieves an average accuracy of 54.94, which is very close to training with the full dataset (55.00).
>
> Budget $b = 4$:
> | Datasets          | AMC23 | MinervaMath | OlympiadBench | MATH500 | AVG   |
> |-------------------|-------|-------------|---------------|---------|-------|
> | NoFinetuning      | 56.04 | 33.90       | 37.28         | 81.03   | 52.06 |
> | RandSelect        | 57.05 | 35.26       | 38.29         | 81.58   | 53.05 |
> | EntSampling       | 56.95 | 35.36       | 38.46         | 81.88   | 53.16 |
> | BADGE             | 54.36 | 34.12       | 37.63         | 80.81   | 51.73 |
> | CEC               | 56.03 | 34.73       | 38.41         | 81.70   | 52.72 |
> | **CONST (ours)**  | **58.21** | **35.83**   | **39.56**     | **82.94** | **54.14** |
> | FullDataset       | 58.70 | 36.66       | 41.04         | 83.61   | 55.00 |
>
> Budget $b = 8$:
> | Datasets          | AMC23 | MinervaMath | OlympiadBench | MATH500 | AVG   |
> |-------------------|-------|-------------|---------------|---------|-------|
> | NoFinetuning      | 56.04 | 33.90       | 37.28         | 81.03   | 52.06 |
> | RandSelect        | 57.48 | 35.91       | 38.77         | 81.90   | 53.52 |
> | EntSampling       | 57.87 | 35.37       | 38.63         | 81.89   | 53.44 |
> | BADGE             | 56.76 | 35.08       | 38.92         | 82.25   | 53.25 |
> | CEC               | 58.28 | 35.21       | 39.18         | 82.16   | 53.71 |
> | **CONST (ours)**  | **59.05** | **36.97**   | **40.19**     | **83.55** | **54.94** |
> | FullDataset       | 58.70 | 36.66       | 41.04         | 83.61   | 55.00 |
>
>
>
> In light of these responses, we hope we have addressed your concerns, and hope you will consider raising your score. If there are any additional notable points of concern that we have not yet addressed, please do not hesitate to share them, and we will promptly attend to those points.

---

### Author Response · Authors · 2025-12-04
**Thank all Reviewers and Area Chairs for your great efforts, insightful comments and support!**

Dear Reviewers and Area Chairs,

We sincerely appreciate your great efforts, insightful comments, support and the constructive suggestions you have provided once again! Through our discussions and the reviewers' responses, it appears that we have effectively addressed the major concerns raised by everyone, and Reviewer H6dA has decided to maintain the high score. This outcome has greatly benefited us, and we would like to express our gratitude to all of you for your support!

The reviewers recognize that the problem addressed by our paper is **important** (Reviewer rXws, Reviewer d26V, Reviewer fBZw), and that our method is **good** (Reviewer rXws), **valuable** (Reviewer H6dA), **novel** (Reviewer d26V), **elegant** (Reviewer d26V), **both efficient and effective** (Reviewer fBZw). The reviewers also recognize that the experiments of our paper are **comprehensive and convincing** (Reviewer H6dA), and that the paper is **cleanly presented** (Reviewer d26V), **easy to follow** (Reviewer H6dA, Reviewer d26V), and **broadly applicable** (Reviewer d26V).

We firmly believe that our framework, CONST, for unsupervised discovery of critical samples in RLVR optimization, plays a significant role in advancing the community. And we are committed to making our complete code and training details publicly available. Moreover, we are eager to engage in further discussions with you to enhance our understanding of the domain and further improve the quality of the paper.

Best,

The Authors

---

### Meta-Review · Area_Chair_cwRr · 2026-01-07

**Summary:**

All reviewers agree the problem—finding a tiny, annotation-free subset that matches full-dataset RLVR performance—is important and timely.
The rebuttal supplied missing baselines (SCF, EWS, log-prob), larger-model results (Qwen-7B), harder-domain tests, calibration-size sensitivity, and a gradient-decomposition justification for the ε-lottery assumption.
Residual doubts: (i) clustering details remain sketchy, (ii) the ε-bound is still heuristic—no empirical ε or gradient proximity shown, (iii) conformal set size may collapse on tiny answer spaces, and (iv) related-work positioning versus core-set/active-learning literature is thin.

**Reviewer Concerns:**

Addressed: Qwen-7B numbers, SCF/EWS baselines, calibration-set sensitivity, clustering recipe, active-loop pilot, harder science tasks, gradient-connection sketch.
Still outstanding: no measurement of actual ε or gradient error; no ablation separating linguistic noise from reasoning uncertainty; limited discussion of broader data-subset-selection literature; clustering ablation is only one alternate K.

**Reviewer Scores:**

rXws: 4 → 4
H6dA: 8 → 8
d26V: 6 → 6
fBZw: 6 → 6

---

### Decision · Program_Chairs · 2026-01-26

Accept (Poster)